# Stereotyped responses of *Drosophila* peptidergic neuronal ensemble depend on downstream neuromodulators

**Wilson Mena[1†], Sören Diegelmann[2‡], Christian Wegener[2], John Ewer[1*]**

[1]Centro Interdisciplinario de Neurociencias de Valparaíso, Universidad de Valparaiso, Valparaiso, Chile; [2]Theodor-Boveri-Institute, University of Würzburg, Würzburg, Germany

**Abstract** Neuropeptides play a key role in the regulation of behaviors and physiological responses including alertness, social recognition, and hunger, yet, their mechanism of action is poorly understood. Here, we focus on the endocrine control ecdysis behavior, which is used by arthropods to shed their cuticle at the end of every molt. Ecdysis is triggered by ETH (Ecdysis triggering hormone), and we show that the response of peptidergic neurons that produce CCAP (crustacean cardioactive peptide), which are key targets of ETH and control the onset of ecdysis behavior, depends fundamentally on the actions of neuropeptides produced by other direct targets of ETH and released in a broad paracrine manner within the CNS; by autocrine influences from the CCAP neurons themselves; and by inhibitory actions mediated by GABA. Our findings provide insights into how this critical insect behavior is controlled and general principles for understanding how neuropeptides organize neuronal activity and behaviors.

*For correspondence: john.ewer@uv.cl

Present address: †Institut de Neurobiologie Alfred Fessard, Gif-sur-Yvette Cedex, France; ‡Department of Neurobiology / Animal Physiology, University of Cologne, Köln, Germany

Competing interests: The authors declare that no competing interests exist.

## Introduction

Understanding how ensembles of neurons produce behaviors is an important aim of neuroscience. The mapping of the neural circuits that underlie a behavior is considered a necessary first step toward this goal, and efforts to determine the 'connectome' of different parts of the nervous system have been present since the beginnings of modern neuroscience. They start with the classical inferred synaptic relationships in Cajal's anatomical analyses (*Ramón y Cajal, 1899*), through the detailed information on the wiring of some invertebrate circuits (e.g., *Carew et al., 1981*; *Comer and Robertson, 2001*; *King, 1976a*, *1976b*), culminating with the complete map of the *Caenorhabditis elegans* central nervous system (CNS) (*White et al., 1986*), and the wiring diagrams of the *Drosophila* optic lobes (*Takemura et al., 2013*) and the mammalian retina (*Helmstaedter et al., 2013*). Yet, research into the functioning of neuronal networks has revealed that a wiring diagram is usually not enough to understand what a neuronal network can do, although it does inform on its possible outcomes. One of the elements that adds tremendous multiplicity to the universe of possible outputs of a neural circuit is the action of neuromodulators, including biogenic amines and neuropeptides. In conjunction with classical transmitters, they can gate the input to a circuit or reconfigure its pattern of activity, thereby causing the same circuit to produce qualitatively different outputs (*Bargmann, 2012*; *Bargmann and Marder, 2013*; *Brezina, 2010*; *Leinwand and Chalasani, 2013*; *Marder, 2012*; *Nusbaum and Blitz, 2012*).

The influence of neuropeptides can be profound and underlies the expression of entire behavioral states, such as hunger and satiation (*Atasoy et al., 2012*; *Chambers et al., 2013*; *Gao and Horvath, 2007*), pair bonding and stress (*Lieberwirth and Wang, 2014*; *Neumann and Landgraf, 2012*), and arousal and attention (*Li et al., 2016*), and can involve many brain regions in addition to sensory and

**eLife digest** Most behaviors occur only under specific circumstances: we eat when we are hungry, for example. But how does the nervous system decide when to start or stop a particular behavior? Molecules called neuropeptides are thought to play a key role in these decisions. Neuropeptides are produced by organs throughout the body and also by the nervous system itself. When neuropeptides act on neurons responsible for a particular behavior – such as feeding – they can inform those neurons about conditions elsewhere in the body and the brain. This enables the nervous system to decide whether to start or stop the behavior. Yet, how the signals from the different neuropeptides are integrated is poorly understood.

As immature insects grow, they regularly molt then shed their outer skeleton – or cuticle – in a process called ecdysis. This requires a series of behaviors to occur in a particular order. The old cuticle is first loosened and shed, and then the new cuticle expands and hardens. A number of neuropeptides control ecdysis: for example, a key neuropeptide called ecdysis-triggering hormone (ETH) triggers the process. However, it was not clear how each of the other neuropeptides that are released at this time contributes to the behaviors involved in ecdysis.

By studying ecdysis in developing fruit flies, Mena et al. now show that the various ecdysial neuropeptides work together to produce the precise behaviors that are observed. For instance, the effect that ETH has on the nervous system depends on whether another neuropeptide called eclosion hormone is also present. ETH can therefore cause different behavioral outcomes depending on the actions of the other neuropeptides.

Further work is needed in order to work out exactly how the nervous system integrates information from different neuropeptides. Do certain neurons respond to specific neuropeptide combinations? It also remains to be seen how different insects are able to use the same neuropeptides to control ecdysis despite their different body shapes.

physiological inputs. How these actions are effected is poorly understood because neuropeptides can be broadly released within the CNS and exert combinatorial and non-linear effects (*Brezina, 2010*). In addition, we know little about how these gatekeepers are themselves regulated, yet such knowledge is also critical for understanding how the expression of a behavioral state is controlled. Here, we report on the genetic dissection, using *Drosophila*, of the response of a network of peptidergic neurons that controls the stereotyped and sequential insect behavior of ecdysis, and which represents a tractable system in which these questions can be addressed.

Ecdysis is the vital behavior that is used by arthropods to shed the remains of the old cuticle at the end of every molt (*Reynolds, 1980*). It includes several behavioral subroutines and physiological events that are expressed sequentially to loosen and then shed the old cuticle, then expand and harden the new one. In insects, it is triggered by the sudden release of the neuropeptide, Ecdysis-Triggering Hormone (ETH) (*Ewer and Reynolds, 2002*; *Zitnan and Adams, 2012*), which activates sequentially a number of peptidergic neurons, each expressing the A isoform of the ETH receptor (ETHR) (*Diao et al., 2016*; *Kim et al., 2006*, *2015*). A current model proposes that each class of peptidergic ETH targets then activates or modulates specific phases of ecdysis behavior (*Kim et al., 2006*). However, the mechanism responsible for producing the sequential activation of these targets in response to a common ETH stimulus is currently unknown.

Here, we combined the use of genetically encoded calcium- and voltage-sensitive probes, targeted RNAi expression, mutants null for particular neuropeptides downstream of ETH, and pharmacology, to understand how this response is produced. For this we focused on the subset of neurons that produce CCAP (Crustacean cardioactive peptide), which play a key role in the control of ecdysis (*Kim et al., 2006*, *2015*; *Lahr et al., 2012*; *Park et al., 2003*). We show that the response of CCAP neurons to ETH and the ensuing ecdysis behaviors depend on direct actions mediated by the ETH trigger as well as on the actions of neuropeptides downstream of ETH together with inhibition mediated by GABA. Importantly, we found that removal of a downstream neuropeptide can eliminate the rhythmic pattern of neuronal activity induced by ETH, revealing that such neuropeptides are critical for the expression of the fundamental features of this neural response. Our findings have

implications for understanding how this vital insect behavior is controlled. The principles that emerge are also relevant for understanding how peptidergic networks control behavioral states.

## Results

### Organization of pupal ecdysial behaviors

At the end of larval life, *Drosophila* enters the prepupal stage, then ecdyses to a pupa to initiate the transformation to the adult that occurs during metamorphosis. Pupal ecdysis consists of a sequence of behavioral subroutines, which starts with the preparatory behavior of pre-ecdysis, during which the hardened larval cuticle of the puparium is loosened from the underlying pupal cuticle through slow anteriorly directed movements of the body. This phase is followed by ecdysis proper, during which alternating left-right contractions lead into a phase of anteriorly directed peristalses that eventually cause head eversion, during which the brain is pushed anterior to the mouth. During the final phase of the behavioral sequence (post ecdysis), alternating left-right contractions and then posteriorly directed movements produce a body with the external shape of an adult fly (*Kim et al., 2006*; *Lahr et al., 2012*; *Park et al., 2003*). Each of these phases has a stereotyped duration and pattern of activity, which can be recorded in intact (*Kim et al., 2015*, *Lahr et al., 2012*; *Park et al., 2003*) and puparium-free preparations (*Kim et al., 2006*; cf., Figure 2, below). In addition, fictive ecdysis can be visualized in ex vivo CNS preparations challenged with ETH that express the calcium sensor, GCaMP, in motoneurons (*Figure 1A,B*). Consistent with the behaviors observed at ecdysis, the pattern of motor activity expressed in vitro in response to ETH consists of an initial phase that primarily recruits activity in the posterior region of the ventral nervous system (VNS) ('P' region, *Figure 1Ab, Ba*; corresponding to pre-ecdysis) followed by a barrage of activity throughout the left and right sides of the VNS neuropils ('L', *Figure 1Ac; Bb*; and 'R' regions, *Figure 1Ad; Bd*). Importantly, expansion of this latter section of the record reveals that the 'L' and 'R' regions are active in an alternating pattern (*Figure 1Be*), consistent with the prominent left-right repetitive contractions seen at ecdysis in the intact animal.

At the endocrine level, ecdysis is initiated by the sudden and near-complete release of ETH from peripheral epitracheal cells, which is fueled by an endocrine positive feedback with centrally produced Eclosion Hormone (EH) (*Ewer et al., 1997*; *Kingan et al., 1997*). In the CNS, direct targets of ETH include neurons that express the neuropeptides, FMRFamide (FMRF), Kinin, EH, and Crustacean Cardioactive Peptide (CCAP), either alone or in combination with bursicon (made up of two subunits, BURS and PBURS) and/or Myoinhibitory Peptides (MIPs) (*Diao et al., 2016*; *Kim et al., 2006*). By targeting a calcium-sensitive GFP (GCaMP) to different peptidergic ensembles, *Kim et al. (2006)* showed that each of these sets of peptidergic neurons is activated at a particular time and for a specific duration following a challenge by ETH in vitro. By correlating these times with the expression of the different behavioral phases of ecdysis, each set was assigned a role in the control of particular ecdysial subroutine, which was also broadly consistent with functional and genetic evidence (*Diao et al., 2016*; *Kim et al., 2006, 2015*; *Krüger et al., 2015*; *Lahr et al., 2012*; *Park et al., 2003*). Nevertheless, the mechanisms by which the direct targets of ETH would be activated at different times after an ETH challenge remained unanswered. Here, we show that the pattern of activity of ETH targets depends on direct ETH actions, as well on actions effected by targets downstream of ETH.

### Role of ETH signaling in the activation of downstream peptidergic targets

In order to understand how the temporal pattern of activity of peptidergic ETH targets is produced, we first identified relevant neuronal sets by determining the effects on ecdysis behavior of expressing ETHR RNAi in each set of peptidergic ETH targets. Expression of ETHR RNAi in FMRFa and EH neurons caused no measurable effect on ecdysis behavior in intact (*Figure 2A*) or puparium-free preparations (*Figure 2B*) ('FMRF>ETHR RNAi' and 'EH>ETHR RNAi', respectively), other than an increase in the frequency of pre-ecdysial contractions. The corresponding pattern of neural activity induced by ETH was not visibly altered by expression of ETHR RNAi in these neurons (not shown). In the case of EH neurons, this lack of effect could be due to their high sensitivity to ETH (*Kim et al., 2006*), which could make them insensitive to the levels of reduction in ETHR expression that can be

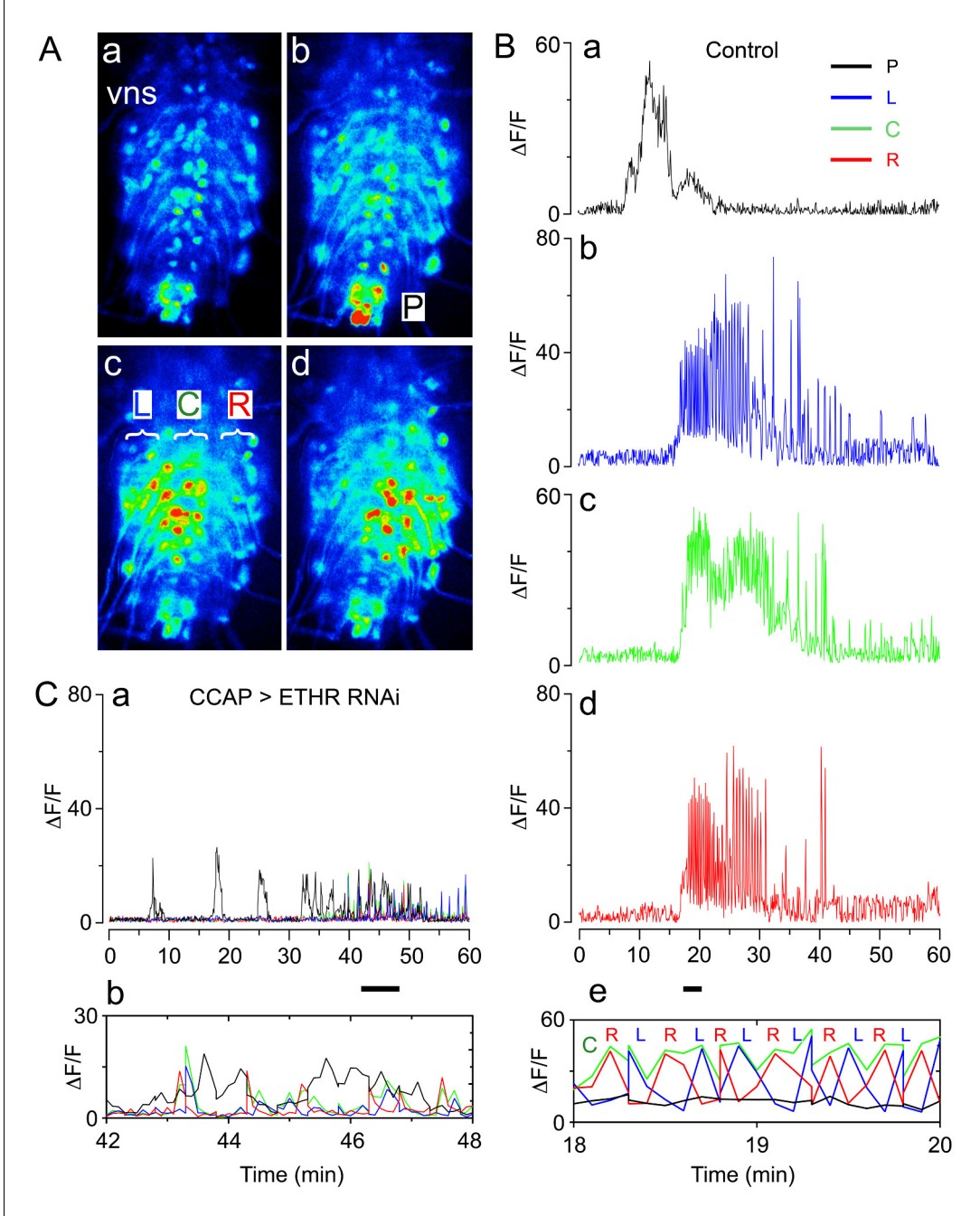

**Figure 1.** Fictive ecdysial behavior in normal and ETHR knockdown animals. (A) Snapshots of the pattern of GCaMP signal recorded from motoneurons of control animals at 0 min (a), 10 min (b), and at around 20 min (c,d) after in vitro challenge with 600 nM ETH. (B) Corresponding recording of GCaMP signal, color-coded according to regions indicated in (Ab and Ac): (a) 'P' (posterior, black trace; cf., Ab); (b): 'L' (left, blue trace; cf., Ac); (c): 'C' (center, green trace; cf., Ac); (d) 'R' (right, red trace; cf., Ac). (e) Expanded segment of recording (recordings for 'P', 'L', 'C' and 'R' regions superimposed) during 18–20 min period (indicated by small bar beneath time axis of (d)). Note the alternating activity in 'L' and 'R' regions. (C) (a) Motoneuron activity patterns (recordings for 'P', 'L', 'C' and 'R' regions superimposed) from animals that express ETHR RNAi in CCAP neurons; (b) Expanded segment of recording shown in (a) (recordings for 'P', 'L', 'C' and 'R' regions superimposed) during 42–48 min period (indicated by small bar beneath time axis in (a)). Note that 'L' and 'R' activity no longer alternate (compare with Be). Zero min indicates time of ETH challenge in all records. Genotypes: Controls (A, B): CCAP>GCaMP (*Ccap*-GAL4 + UAS-GCaMP); ETHR knockdown in CCAP neurons (C): CCAP+MN>GCaMP+ETHR RNAi (MN: C164 motoneuron GAL4; see Materials and methods). Note that this genotype also knocks down ETHR expression in motoneurons (MNs). Nevertheless, knockdown of ETHR only in MNs had only a slight effect on ecdysis behavior (cf. *Figure 2*), suggesting that most defects observed here were due to knockdown of ETHR in CCAP neurons. In all experiments using RNAi, its effectiveness was boosted by including a UAS-*dcr2* transgene.

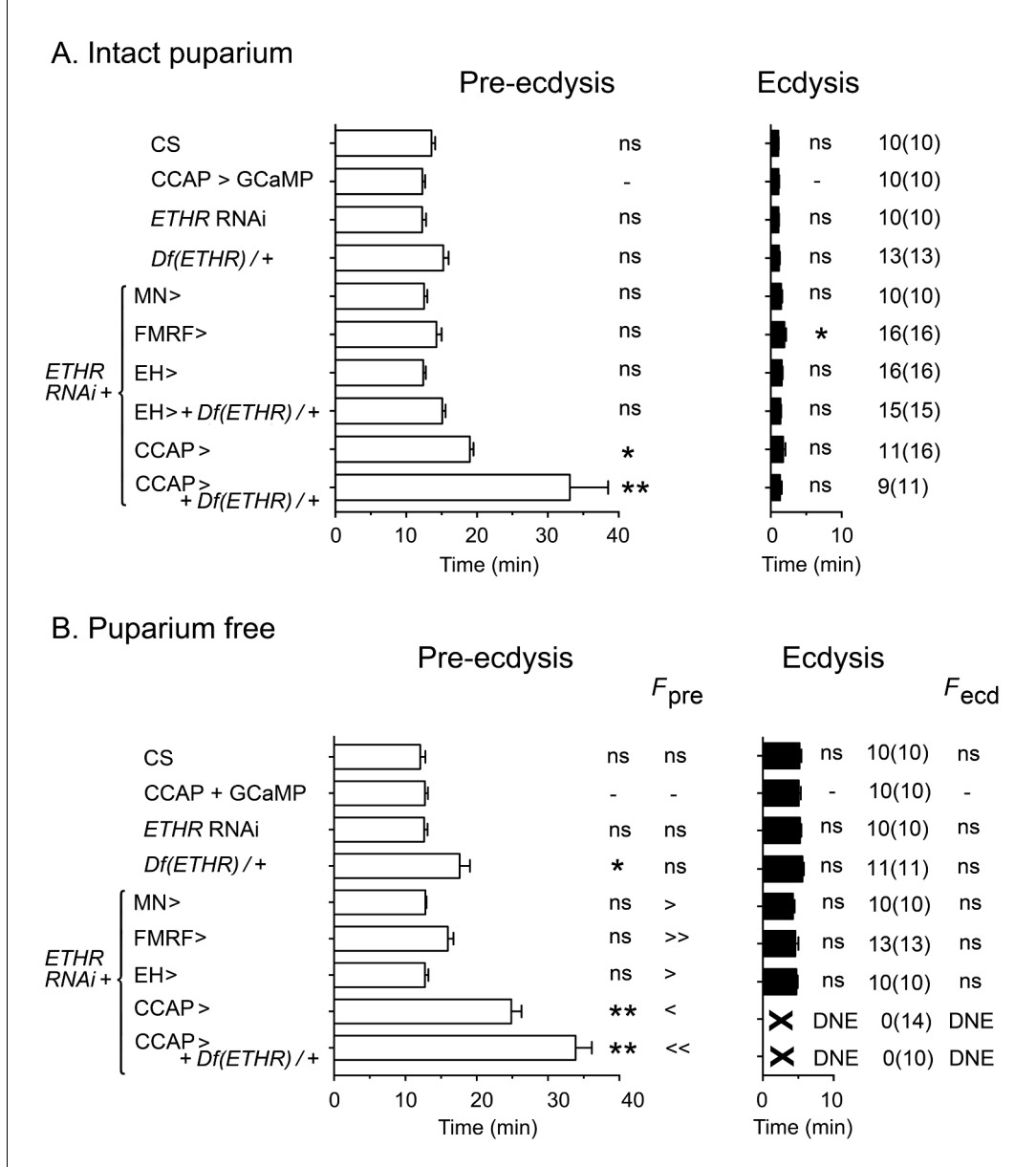

**Figure 2.** Impact on ecdysial behaviors of manipulating ETH effectiveness on downstream targets. (**A**) Ecdysial behaviors in intact puparium. Duration of pre-ecdysis (left; open bars) and ecdysis (right; filled bars) in controls (CS and CCAP>GCaMP), and in animals expressing ETHR RNAi in different subsets of ETH targets. (**B**) Corresponding ecdysial behaviors of puparium-free preparations. Data are mean ± SEM. In (**A**) and (**B**) duration of ecdysial phases is indicated as not significantly different ('ns') or significantly different ('*': p<0.5; '**': p<0.01) than those of CCAP>GCaMP control (one-way ANOVA, Dunnett's *post-hoc* to control). Comparison of frequency of contractions per minute for pre-ecdysis and ecdysis (labeled $F_{pre}$ and $F_{ecd}$, respectively) in puparium-free preparations is indicated as not significantly different ('ns') or significantly smaller ('<': p<0.05; '<<' p<001) than or greater ('>': p<0.05; '>>' p<001) than that of CCAP>GCaMP controls (one-way ANOVA, Dunnett's *post-hoc* to control). Actual p values can be found in **Supplementary file 1**. Genotypes: all animals expressed GCaMP under control of *Ccap*-GAL4 (*Ccap*-GAL4 + UAS-GCaMP). ETHR RNAi: UAS-ETHR RNAi; *Df(ETHR)/+*: hemizygosity for ETHR; MN>: motoneuron GAL4 (C164; see Materials and methods). In all experiments using RNAi, its effectiveness was boosted by including a UAS-*dcr2* transgene.

achieved by RNAi (see below). Recent work has shown that the role of kinin neurons in the control of ecdysis behavior (**Kim et al., 2015**) may be mostly indirect and a consequence of its role in diuresis (**Diao et al., 2016**).

By contrast, expression of ETHR RNAi in CCAP neurons had a considerable effect on ecdysis behavior (*Figure 2*; *Kim et al., 2015*). In particular, animals monitored free of the puparium expressed a significantly longer and weaker pre-ecdysis than did controls, and none (0/14) ecdysed (*Figure 2B*). The defects in pre-ecdysis were rendered more severe in animals hemizygous for *ETHR* (*Figure 2*), revealing that the phenotype obtained by knockdown of ETHR using ETHR RNAi is functionally equivalent to that of an *ETHR* hypomorph. Fictive ecdysis behaviors, monitored by expressing GCaMP in motor neurons, showed that the underlying motor program was also severely disrupted in these animals, consistent with the expressed behavior. Indeed, unlike the sequences of calcium responses observed in the motoneurons of control animals (*Figure 1A,B*), those of animals expressing ETHR RNAi in CCAP neurons responded to ETH with a smaller and sustained pattern of activity in the posterior region (*Figure 1C*), consistent with the longer and weaker pre-ecdysis behavior observed (*Figure 2*), and a severe reduction and disorganization in the activity of L and R regions of the CNS, consistent with the observed lack of ecdysis behavior (*Figure 2*). These results confirm previous findings (*Kim et al., 2006*, *2015*; *Lahr et al., 2012*; *Park et al., 2003*) and reveal that CCAP neurons play a critical role in the control of *Drosophila* ecdysis behavior.

Knockdown of ETHR in CCAP neurons also caused significant changes in the calcium response induced by ETH in vitro. In control animals, the two pairs of serial CCAP homologs showed a characteristic response to 600 nM ETH: whereas one set of neurons (called here α neurons, *Figure 3C–E*) responded ca. 20 min after ETH challenge with a barrage of GCaMP spikes, the other (called here β neurons, *Figure 3C,D*) responded after a similar delay with a sustained increase in fluorescence. Based on the electrophysiological activity recorded in *Manduca* (*Gammie and Truman, 1997*), we assume that the α neurons correspond to the projection 'Cell 27' neurons, whereas the β neurons correspond to interneurons 'IN704'. Expression of ETHR RNAi in CCAP neurons caused a significant delay in the onset of the response (*Figure 3F,H*; *Kim et al., 2015*), and the period of activity was shorter and included a smaller number of peaks (*Figure 3F,I,J*) (In these records, the responses of α and β neurons have been combined as they could not be unambiguously distinguished). This weakening of the response was seen in CCAP neurons in thoracic segment 3 (TN3) as well as in serial homologs in abdominal segments 1–4 (AN1-4) but was not detected in AN8-9, which may not express ETHR (*Diao et al., 2016*; *Kim et al., 2006*). Further reduction of ETHR function, accomplished using *ETHR* hemizygosity, did not simply reduce the response further, revealing that indirect as well as direct ETH actions may be involved in determining the response of CCAP neurons to ETH. Indeed, in such animals, the latency to respond was generally more similar to that of the control (*Figure 3G,H*), whereas its duration and number of spikes was more similar to those observed when ETHR RNAi was expressed in a wild-type background (*Figure 3I,J*). Significantly, however, as illustrated in *Figure 3F,G* and quantitated in *Figure 3—figure supplement 1*, both manipulations severely reduced the amplitude of the response. Thus, for animals expressing ETHR RNAi it was reduced four-fold for CCAP neurons AN1-4 and AN8-9, and eight-fold in CCAP neurons TN3, AN1-4, and AN8-9 for animals expressing ETHR RNAi in a ETHR hemizygous background. This reduced response may be the basis for the much weaker behavior expressed by these genotypes (*Figures 1C* and *2*). We do not know the cause of the differential effects on CCAP neurons TN3 vs. AN1-4 for these and other manipulations carried out in this study; we assume that they are due to differential ETHR expression and/or synaptic inputs.

As an alternative approach to studying the response of CCAP neurons to ETH by changing their sensitivity to ETH, we investigated the effects of challenging the CNS with lower doses of ETH. As was observed when ETHR signaling was reduced using RNAi, such manipulations again revealed the presence of non-linear effects. Thus, for instance the lengthening of the latency observed using 300 nM was in general reversed with the lower concentrations (150 nM and 60 nM; *Figure 4E*). Interestingly, none of these concentrations significantly affected the duration of the response (*Figure 4F*). Nevertheless, the responsiveness of neurons was severely affected, with only around 75%, 50% and 25% of neurons responding when challenged with 300 nM, 150 nM and 60 nM ETH1, respectively (vs. 100% for 600 nM ETH1). Furthermore, the amplitude of the response was significantly decreased for all the lower concentrations used (ten- to forty-fold, depending on segmental location and ETH concentration; *Figure 4B–D*-Figure; see also *Figure 3—figure supplement 1*).

These results show that the response of CCAP neurons to ETH affects the expression of ecdysis behaviors and that this response depends in a nonlinear fashion on the levels of both ligand and

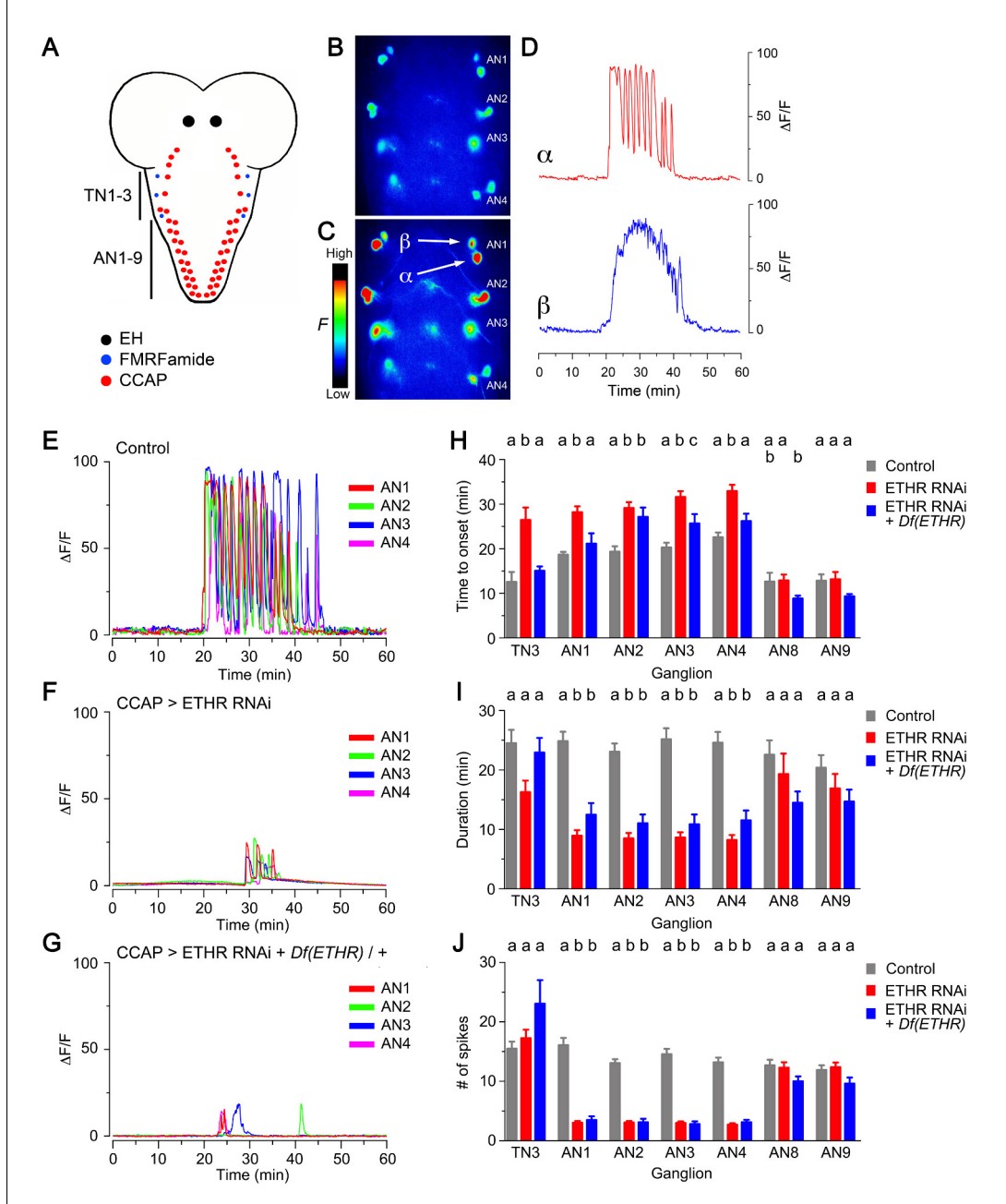

**Figure 3.** Impact on CCAP neuron activation of ETHR knockdown in CCAP neurons. (A) Schematic of *Drosophila* nervous system indicating the location of EH, FMRFa, and CCAP neurons. (B–C) Snapshots of the pattern of GCaMP signal in CCAP neurons in segments AN1-4, recorded from wild-type animals at 0 min (B) and 20 min (C) after in vitro challenge with 600 nM ETH. (D) Calcium dynamics of AN1 α (top, red trace) and β (lower, blue trace) neurons (cf., 3C) after ETH challenge. (E–G) Pattern of GCaMP activity recorded from CCAP neurons AN1-4 following in vitro challenge with 600 nM ETH in wildtype CNSs (E), in CNSs expressing ETHR RNAi in CCAP neurons (F), and in CNSs of *ETHR* hemizygous animals expressing ETHR RNAi in CCAP neurons (G). (H–J) Quantitation of time of onset (H), duration (I), and number of spikes (J) for the different genotypes tested. TN3: thoracic ganglion 3; AN: abdominal ganglion. Zero min indicates time of ETH challenge. N = 10 preparations for all genotypes. Data are mean ± SEM. Different letters indicate statistically significant groups (p<0.05); one-way ANOVA, Tukey's *post-hoc* multiple comparison analyses. Actual p values can be found in *Supplementary file 1*. Genotypes: all animals expressed GCaMP under control of *Ccap*-GAL4 (*Ccap*-GAL4 + UAS-GCaMP); ETHR RNAi: UAS-ETHR RNAi; *Df(ETHR)/+*: hemizygosity for ETHR. In all experiments using RNAi, its effectiveness was boosted by including a UAS-*dcr2* transgene.

The following figure supplement is available for figure 3:

**Figure supplement 1.** Amplitude of GCaMP response induced by ETH in CCAP neurons.

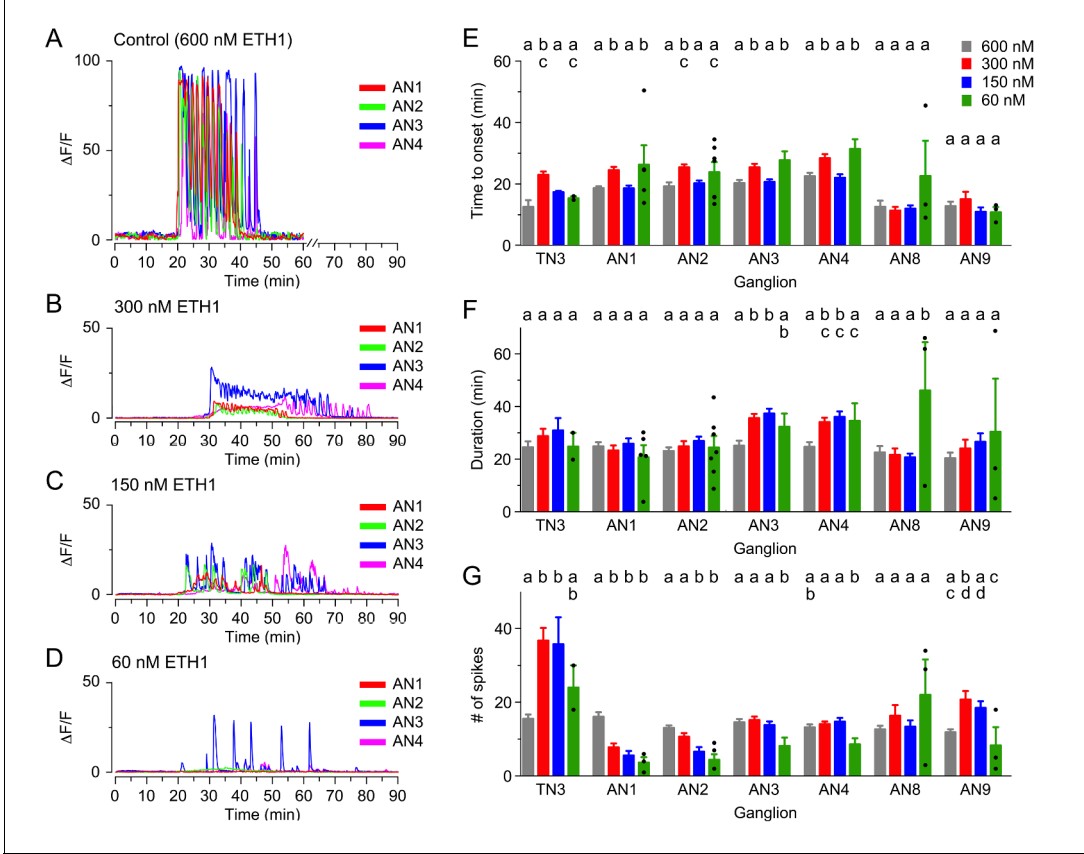

**Figure 4.** Calcium responses induced by different concentrations of ETH in CCAP neurons. (**A–D**) Pattern of GCaMP activity recorded from CCAP neurons in AN1-4 from control CNSs challenged in vitro with 600 nM (**A**), 300 nM (**B**), 150 nM (**C**), and 60 nM (**D**) ETH. Zero min indicates time of ETH challenge. (**E–G**) Quantitation of results, shown as described in *Figure 3H–J*. N = 10 for all preparations. Data are mean ± SEM. Different letters indicate statistically significant groups (p<0.05); one-way ANOVA, Tukey's *post-hoc* multiple comparison analyses. Actual p values can be found in *Supplementary file 1*. The responsiveness and amplitude of the responses were also affected; see text and *Figure 3—figure supplement 1*. Individual measurements have been superimposed on the summary histogram when <8 neurons showed a measurable response. Genotypes: all animals expressed GCaMP under control of *Ccap*-GAL4 (*Ccap*-GAL4 + UAS-GCaMP).

receptor. This non-linearity could simply reflect the sigmoidal dose-response curve of GPCRs or may also be caused by inputs to CCAP neurons that are also (direct or indirect) targets of ETH.

## Influence of downstream peptides on the pattern of activation of CCAP neurons by ETH: role of eclosion hormone

Eclosion hormone (EH) is a key neuropeptide in the control of ecdysis (*Ewer and Reynolds, 2002*; *Gammie and Truman, 1999*; *Krüger et al., 2015*). It is involved in potentiating and accelerating the release of ETH that causes the sudden onset of ecdysis behaviors (*Ewer et al., 1997*; *Kingan et al., 1997*); in addition, it plays a critical role within the CNS for the expression of ecdysis behaviors. In *Drosophila* larvae, for instance, mutants lacking EH do not express pre-ecdysis, transitioning directly into a prolonged ecdysis-like phase, which is, nevertheless, generally ineffective in causing the shedding of the old cuticle (*Krüger et al., 2015*). *Eh* mutant animals also expressed specific behavioral defects at pupal ecdysis, which, interestingly, differed from those expressed in the larva. Thus, intact *Eh* hemizygous mutants expressed much longer pupal pre-ecdysis (*Figure 5A*) and ecdysis behavior was also longer, with most animals (4/10) failing to express the behavior. Similar defects were observed in puparium-free preparations (*Figure 5B*), although in this case none (0/10) expressed the ecdysial phase of the behavior. These defects were all qualitatively rescued using a transgene containing the *Eh* gene, indicating that they were mostly due to the lack of EH (*Figure 5A,B*, for intact and puparium-free preparations, respectively).

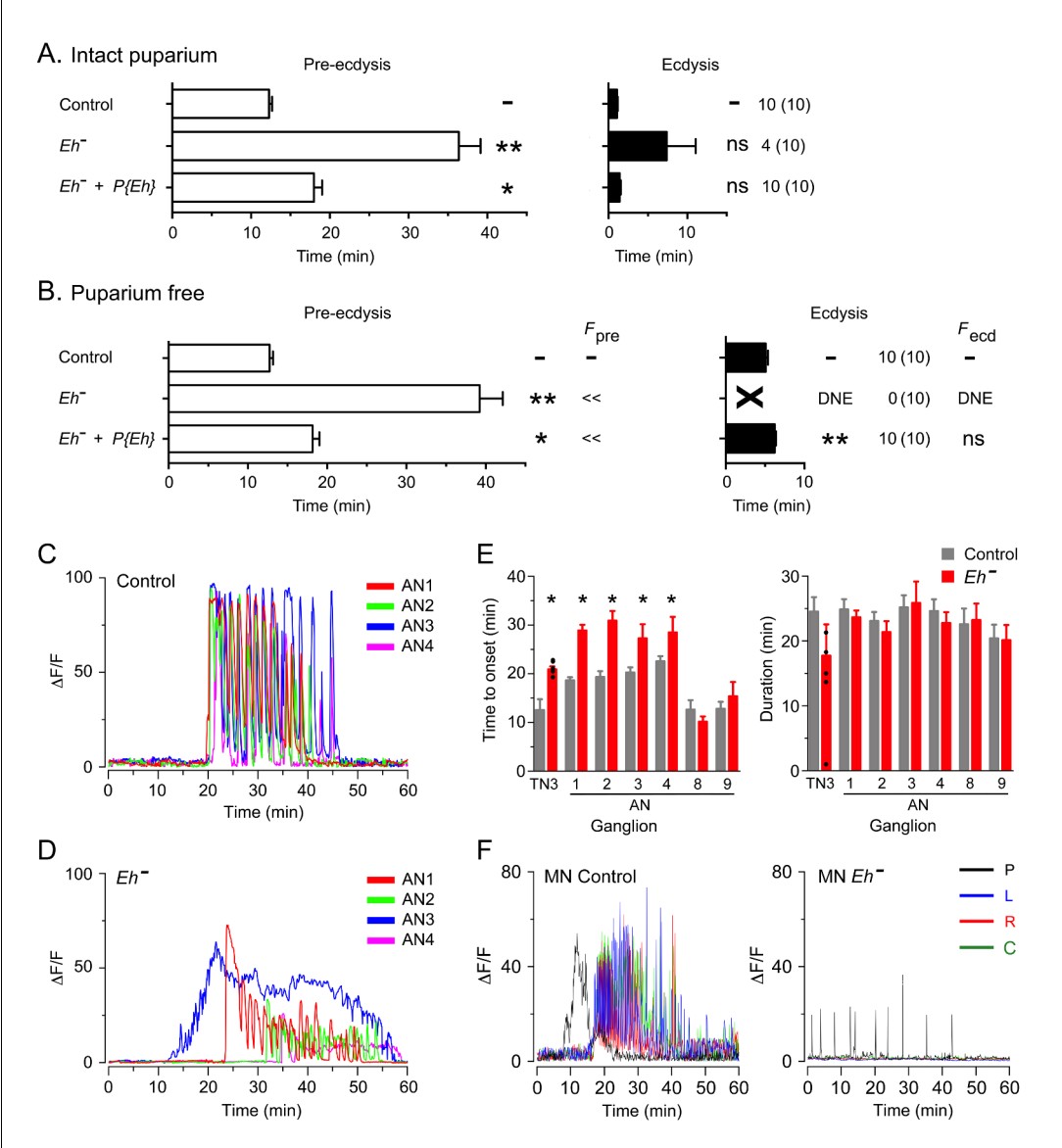

**Figure 5.** Absence of EH abolishes ecdysis and alters the response of CCAP neurons to ETH. (A,B) Duration of pre-ecdysis (left; open bars) and ecdysis (right; filled bars) behavior of animals hemizygous for *Eh* in intact (A) and puparium-free preparations (B), shown as described in *Figure 2*. Behavioral defects in (A, B) were qualitatively rescued by transgene containing *Eh* gene, indicating that they are caused by lack of EH neuropeptide. (C,D) Pattern of GCaMP activity in CCAP neurons AN1-4 induced by ETH in CNS from control animals (C) and from animals hemizygous mutant for *Eh* (D). Zero min indicates time of ETH challenge. (E) Summary of results obtained for latency (left) and duration (right) of response. (F) Recording from motoneurons from CNS of controls (left; cf. *Figure 1B*) and animals hemizygous for *Eh* (right); traces color-coded as described in *Figure 1*. Zero min indicates time of ETH challenge. N = 6–10 for all genotypes. Data in (A, B, and E) are mean ± SEM. One-way ANOVA, Dunnett's *post-hoc* to control; (*: p<0.05; **: p<0.01); actual p values can be found in *Supplementary file 1*. The amplitude of the responses was also affected; see text and *Figure 3—figure supplement 1*. In (E), individual measurements have been superimposed on the summary histogram when < 8 neurons showed a measurable response. Genotypes: in (A–D) all animals expressed GCaMP under control of *Ccap*-GAL4; in (F), they expressed GCaMP under control of MN-GAL4 (C164-GAL4). *Eh[-]: Eh[-]/Df(3)Eh; Eh[-]+P{Eh}: Eh[-]/Df(3)Eh; P{Eh}*; see Materials and methods for exact genotypes.

*Drosophila* EH neurons are direct targets of ETH (*Diao et al., 2016*; *Kim et al., 2006*) and respond to ETH with a shorter latency than do CCAP neurons (*Kim et al., 2006*); in addition, the response of larval CCAP neurons to ETH is severely weakened in the absence of EH (*Krüger et al., 2015*). These results suggest that EH neurons are an ETH target that is upstream of CCAP neurons. In order to explore the role of EH in determining the features of the response of CCAP neurons to

ETH at pupal ecdysis, we examined the GCaMP signal of CCAP neurons to ETH in animals hemizygous mutant for *Eh*. As shown in *Figure 5D*, the temporal organization of the response was dramatically disorganized in these animals, with a significant loss in the regular pattern of GCaMP spikes that is typical of the control (*Figure 5C*). This effect was partially reflected in the quantitation of the response, where the lack of EH caused significant increases in the latency to respond (*Figure 5E*) and a two-fold decrease in the amplitude of the response in CCAP neurons AN1-4 and AN8,9 (*Figure 5D*; see also *Figure 3—figure supplement 1*), although the severely disrupted response (cf., *Figure 5D*) precludes a quantitative comparison. These results show that, as occurs in the larva, the response of CCAP neurons to ETH depends critically on the action of EH.

Animals hemizygous mutant for *Eh* also expressed a severely weakened and disrupted pattern of motoneuron activity in response to ETH (*Figure 5F*). This aberrant response is consistent with the failure of ecdysis behavior observed in this genotype (*Figure 5A,B*), although in the intact animal the absence of EH would likely also prevent ETH release (*Krüger et al., 2015*), which may also contribute to the failure of ecdysis,

## Influence of downstream peptides on the pattern of activation of ETH targets: role of PBURS

Bursicon, the so-called tanning hormone, has traditionally been implicated in post-ecdysial events following adult emergence (*Honegger et al., 2008*). However, genetic analyses have shown that *pburs*, which encodes one of the subunits of the heterodimeric bursicon hormone, also plays a role in the control of ecdysial behaviors (*Lahr et al., 2012*). Bursicon is of particular interest in the context of this work because it is expressed by a subset of the CCAP neurons themselves and could therefore play a (direct or indirect) autocrine role in determining the response of these neurons to ETH. Consistent with the findings of *Lahr et al. (2012)*, the lack of PBURS (and thus of the heteromeric bursicon) in intact puparia caused a lengthening of ecdysis, with only 5/8 animals executing the behavior (*Figure 6A*). Puparium-free preparations lacking PBURS, by contrast, expressed a significantly shorter ecdysis motor program (*Figure 6B*); a similar defect was obtained expressing *pburs* RNAi in CCAP neurons (*Figure 6B*). Importantly, the defects expressed by hemizygous mutant animals in both types of preparations were rescued by a transgene containing the *pburs* gene (*Figure 6A,B*) indicating that they are due to the lack of *pburs* function. The apparently opposite phenotypes expressed by these two types of preparations are likely due to the different criteria that are used to define the ecdysial phases in intact vs. puparium-free animals. At the level of the response of CCAP neurons, the lack of PBURS caused, on average, a two-fold decrease in the amplitude of the response to ETH of CCAP neurons AN1-4 (*Figure 7B*; see also *Figure 3—figure supplement 1*). (These and other experiments involving mutations in *pburs* had to be done using the LexA/LexAop binary expression system due to interference from GAL4; see legend to *Figure 6* for further details.) It also accelerated the time of onset and shortened the response for some of the CCAP neurons (*Figure 7C,D*). No effects on the timing or amplitude of the response were detected in CCAP neurons AN8-9, suggesting that they may not express the bursicon receptor. In summary, the absence of *pburs* function caused quantitative changes to the response of CCAP neurons to ETH, revealing that a neurohormone made by the CCAP neurons themselves participates in determining their response to the ETH trigger.

## Inhibitory influences in the activation of ecdysis sequence

A striking feature of the responses triggered by ETH on its peptidergic targets is that, despite all being direct targets of this triggering hormone, the various subsets of peptidergic neurons are not activated simultaneously following an in vitro ETH challenge. Thus, although EH (*Kim et al., 2006*) and Kinin (*Diao et al., 2016*; *Kim et al., 2015*) neurons respond within ca. 10 min of an ETH challenge, CCAP neurons do not respond until around ca. 20 min after ETH stimulation, with different subsets showing different and characteristic latencies (*Figure 8C,F*; *Kim et al., 2006*). The delay in the execution of the ecdysial phase of ecdysis behavior, which correlates with the time of activation of CCAP neurons, has been hypothesized to occur through inhibitory influences originating from the brain and/or subesophageal ganglion (*Baker et al., 1999*; *Ewer and Reynolds, 2002*; *Ewer and Truman, 1997*; *Fuse and Truman, 2002*; *Zitnan and Adams, 2000*). Nevertheless, the exact origin and nature of this inhibition is currently unknown. In order to investigate the role of inhibition in the

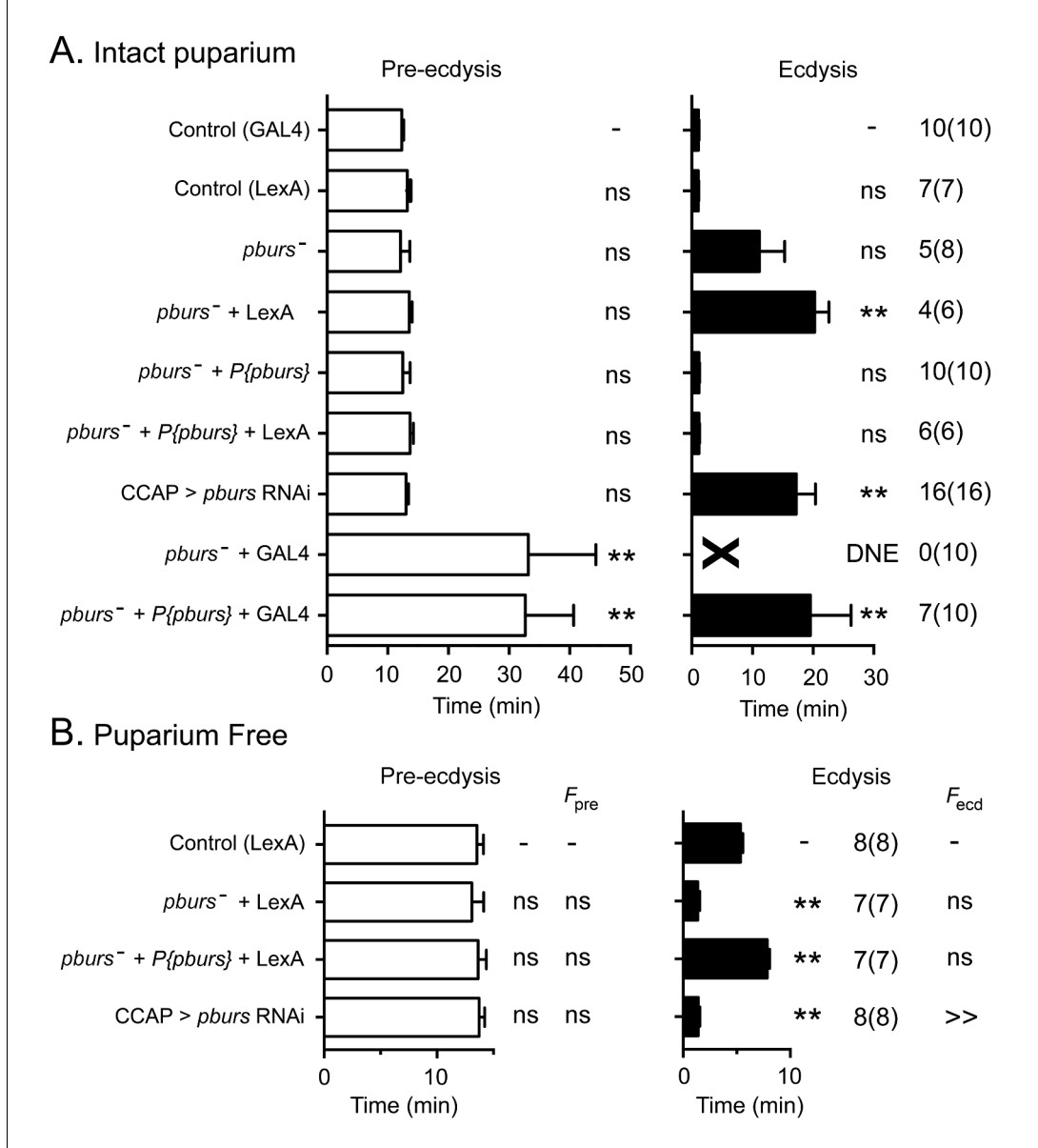

**Figure 6.** Ecdysial behaviors of *pburs* mutant animals. (**A,B**) Duration of pre-ecdysis (left; open bars) and ecdysis (right; filled bars) behavior of animals hemizygous for *pburs* in intact (**A**) and in puparium-free preparations (**B**), shown as described in *Figure 2*. Data are mean ± SEM; one-way ANOVA, Dunnett's *post-hoc* to control. Actual p values can be found in *Supplementary file 1*. Lack of *pburs* function did not significantly alter the duration of the pre-edysis phase (unless CCAP neurons also expressed GAL4, see below), whereas it lengthened (intact preparations) or shortened (puparium free preparations) the duration of ecdysis; a similar result was obtained when *pburs* function was knocked down using RNAi. In (**A**), note that expression of GAL4 in CCAP neurons accentuated the defects expressed by *pburs* hemizygotes; these defects could not be rescued by *pburs*-containing transgene and were therefore not caused only by the lack of *pburs* function. For this reason, GCaMP responses to ETH of *pburs* mutants were assessed using *Ccap*-LexA driver (cf., *Figure 7*). Genotypes: Control (GAL4): CCAP-GAL4+UAS-GCaMP; Control (LexA): CCAP-LexA+LexAop-GCaMP. All genotypes including '(GAL4)' or ('LexA') contained CCAP-GAL4+UAS-GCaMP or CCAP-LexA+LexAop-GCaMP, respectively. *pburs[-]: pburs[-]/Df(2)pburs; pburs*[-] +P{*pburs*}: *pburs[-]/Df(2)pburs*; P{*pburs*}; see Materials and methods for exact genotypes. In all experiments using RNAi, its effectiveness was boosted by including a UAS-*dcr2* transgene.

response of CCAP neurons to ETH, we first examined their time course of activation in the presence of the GABA$_A$ receptor (GABA-RA) blocker, picrotoxin (100 μM; *Rohrbough and Broadie, 2002*). As shown in *Figure 8D,F*, pharmacological inhibition of GABA-RA caused a significant reduction in the latency to respond; this effect was most dramatic in neurons from segments AN1-4, where the time

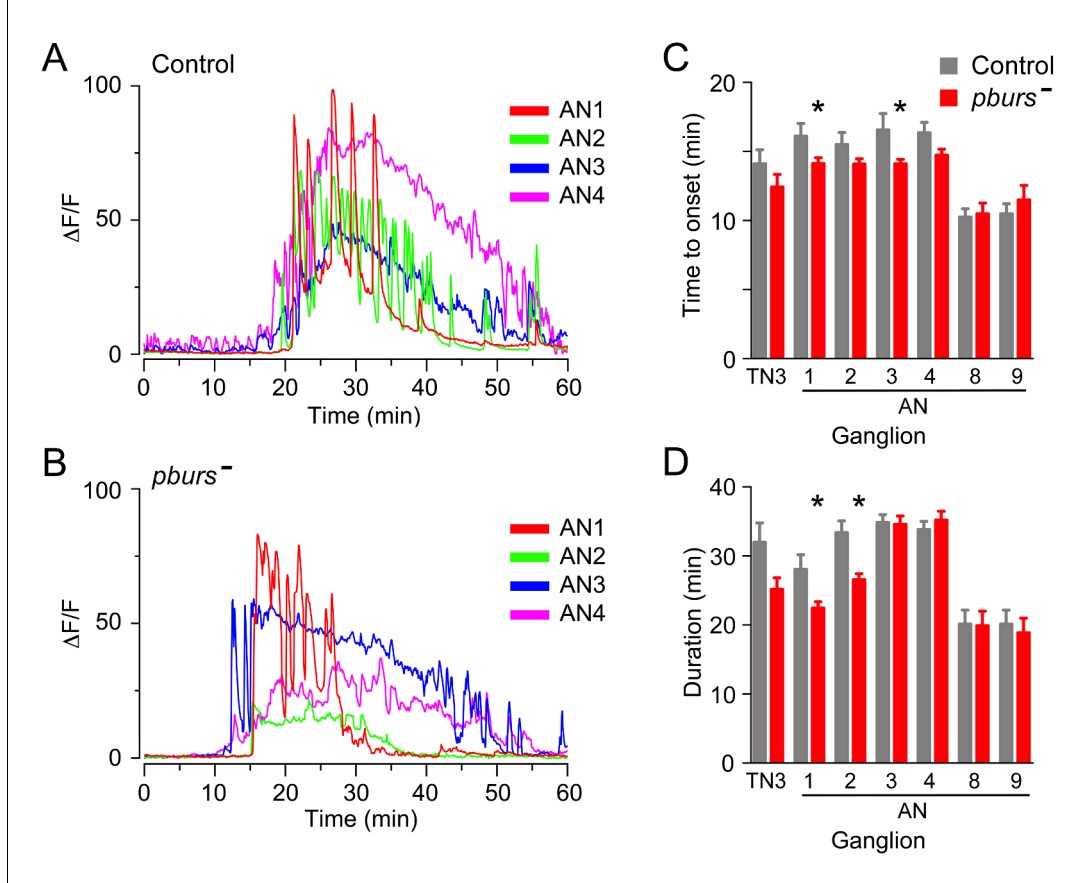

**Figure 7.** Absence of PBURS affects response of CCAP neurons to ETH. (**A,B**) Pattern of GCaMP activity in CCAP neurons AN1-4 induced by ETH in CNS from control animals (**A**) and in CNS of animals hemizygous mutant for *pburs* (**B**). Zero min indicates time of ETH challenge. (**C,D**) Summary of results obtained for latency (**C**) and duration (**D**) of response, summarized as described in *Figure 5*. N = 7–10 for all genotypes and preparations. Data in (**C,D**) are mean ± SEM. Significant differences (p<0.05) compared to control are indicated by '*'; *t* test (unpaired, two-tailed). Actual p values can be found in *Supplementary file 1*. The amplitude of the responses was also affected; see text and *Figure 3—figure supplement 1*. Genotypes: Control: CCAP-LexA+LexAop-GCaMP. *pburs[-]: pburs[-]/Df(2)pburs* + CCAP-LexA+LexAop-GCaMP; see Materials and methods for exact genotypes.

to onset was reduced by almost 50% (from around 20 to 10 min). Other changes were also apparent in these preparations. Most noteworthy was the more sustained nature of the response (*Figure 8D*), which suggests that the pronounced spikes that are normally seen (e.g., *Figure 8C*) are partly accomplished through inhibitory processes mediated by GABA. Similar results were obtained using the GABA-RA inhibitor, gabazine (100 µM; *Hosie and Sattelle, 1996*; data not shown).

In order to determine the contribution of inhibitory processes acting directly on CCAP neurons, we expressed GABA-RA RNAi in these ETH targets. As illustrated in *Figure 8E* and quantitated in *Figure 8F*, this manipulation caused a significant shortening of the response latency, which, furthermore, was similar to that observed following picrotoxin inhibition (*Figure 8D,F*), but without significantly affecting the other features of the response (duration and number of spikes, *Figure 8G and H*, respectively). Importantly, the corresponding animals expressed a significantly shorter pre-ecdysis, thereby reducing the latency to ecdysis (*Figure 8A,B*). These results show that around 50% of the latency of the response of CCAP neurons to ETH is caused by GABA inhibition acting directly on these neurons. They also again show the importance of CCAP neurons in the control of ecdysis since reducing the latency of the onset of the GCaMP response was accompanied by a comparable shortening in the time of onset of ecdysis.

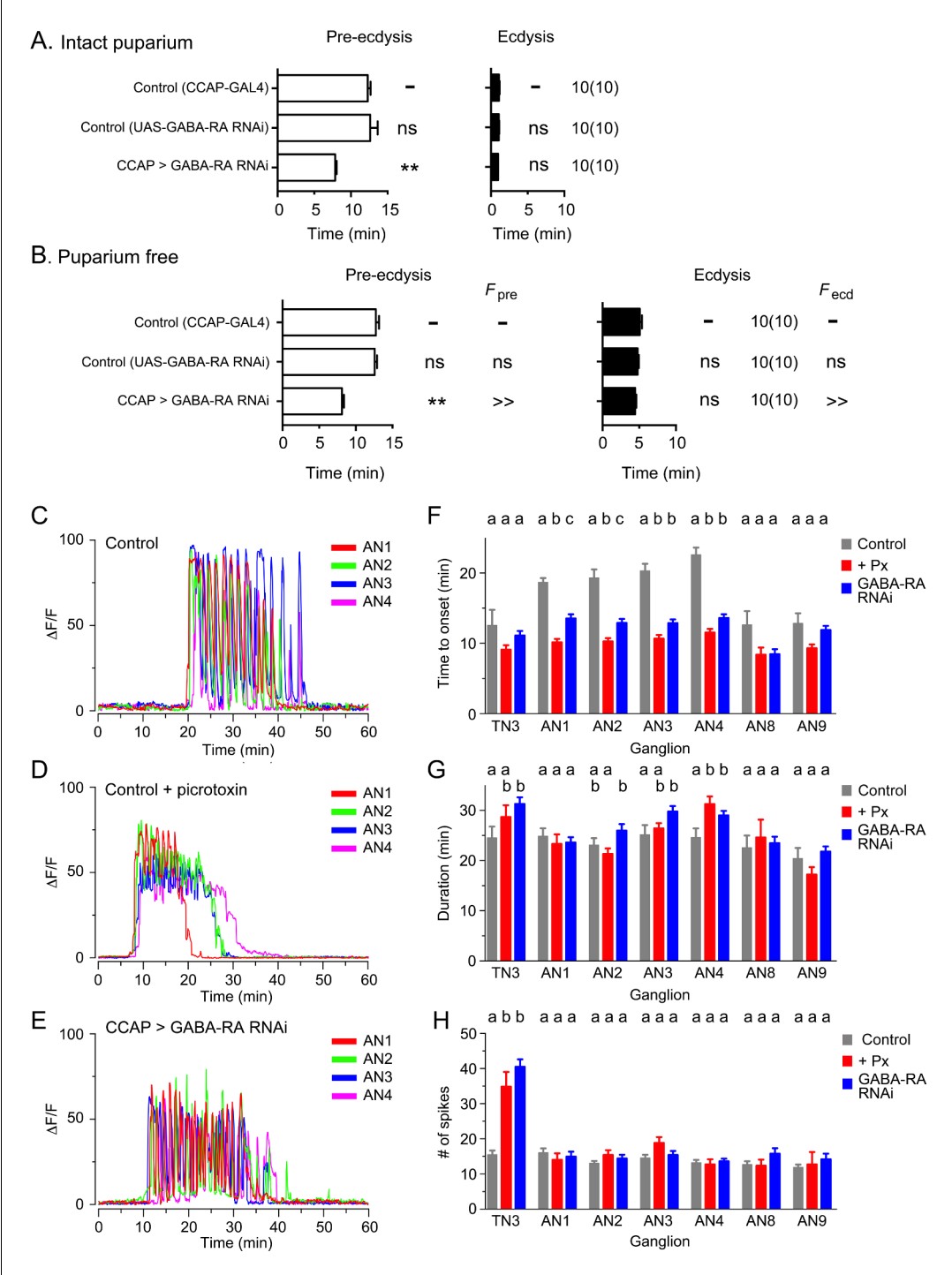

**Figure 8.** GABA inhibition controls latency to ecdysis and of CCAP response to ETH. (**A,B**) Duration of pre-ecdysis (left; open bars) and ecdysis (right; filled bars) behavior in intact (**A**) and in puparium-free (**B**) preparations expressing GABA-RA RNAi in CCAP neurons; results are summarized as shown in *Figure 2*. Animals also expressed GCaMP under control of *Ccap*-GAL4. (**C–E**) Pattern of GCaMP activity in CCAP neurons AN1-4 induced by ETH in CNSs from control animals (**C**), in CNSs from control animals recorded in the presence of GABA-RA antagonist picrotoxin (100 μM) (**E**), and in CNSs expressing GABA-RA RNAi in CCAP neurons (**E**). (**F–H**) Quantitation of results, shown as described in *Figure 3H–J*. Zero min indicates time of ETH challenge. N = 9–13 for all genotypes and preparations. Data in (**A,B**, **F–H**) are mean ± SEM. For panels **A** and **B**, data for experimental groups were compared to those of control (one-way ANOVA, Dunnett's *post-hoc* to control), and summarized as described in *Figure 2*. For panels **F–H**, different letters indicate statistically significant groups (p<0.05); one-way ANOVA, Tukey's *post-hoc* multiple comparison analyses. Actual p values for all analyses

*Figure 8 continued on next page*

*Figure 8 continued*
can be found in **Supplementary file 1**. All animals also expressed GCaMP under control of *Ccap*-GAL4 (*Ccap*-GAL4 + UAS-GCaMP). In all experiments using RNAi, its effectiveness was boosted by including a UAS-*dcr2* transgene.

## Direct measurement of GABA-mediated inhibition

In order to more directly visualize the inhibitory processes acting on CCAP neurons, we carried out optical voltage recordings of these neurons using the genetically encoded voltage sensor, ArcLight (*Cao et al., 2013*). As shown in **Figure 9B**, two different responses could be recorded in preparations where an α + β pair of CCAP neurons was in focus (neurons from neuromeres A8 and A9 could not be visualized so are omitted from these analyses): although both neurons showed an initial plateau (top red and blue records in **Figure 9B**) followed by a depolarization at around 20 min (black

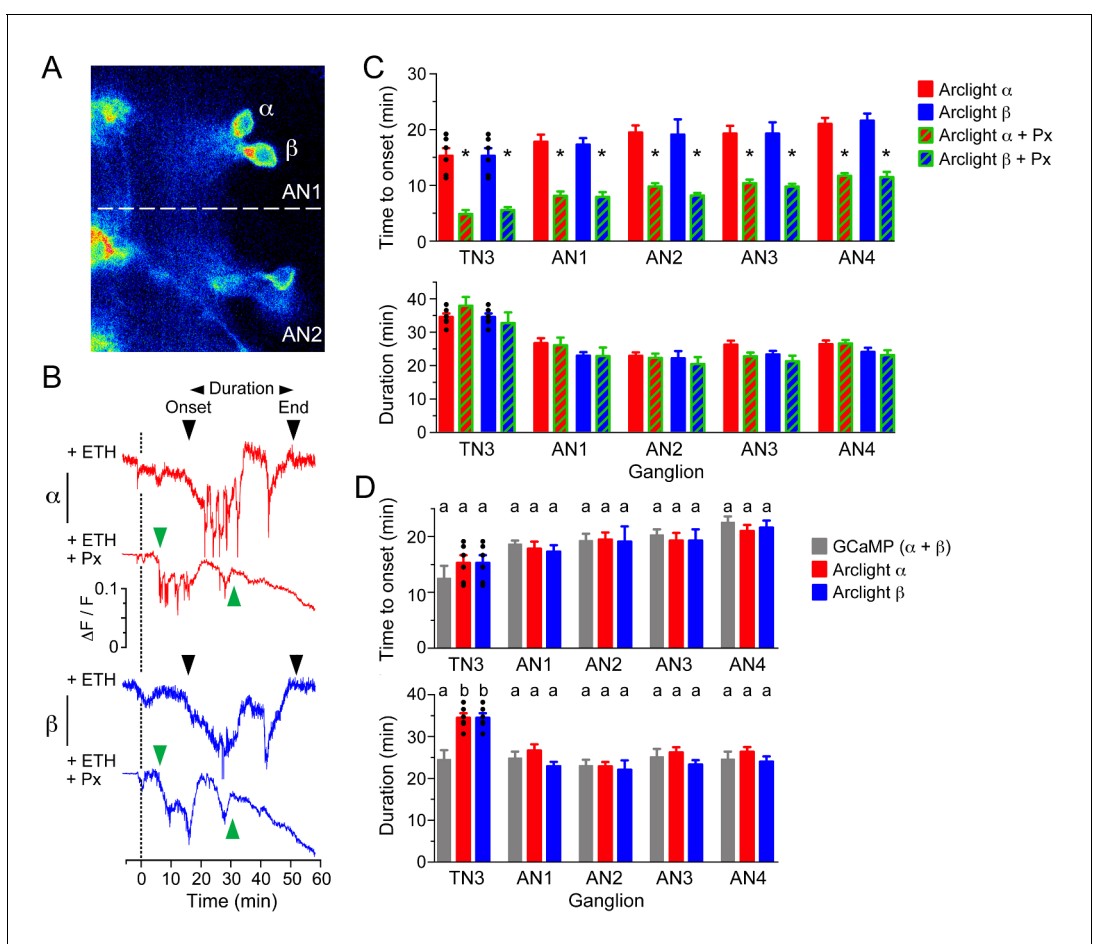

**Figure 9.** GABA delays depolarization of CCAP neurons induced by ETH. (**A**) ArcLight fluorescence in CCAP neurons α and β from ganglion AN1-2. (**B**) ArcLight signal recorded from α (red trace) and β (blue trace) neurons following ETH stimulation in control CNS (+ETH; top traces) and in the presence of picrotoxin (+ETH+Px, lower traces). Zero min indicates time of ETH challenge. Inverted triangles indicate start and end of depolarization; 'Duration' corresponds to the time between these two events. (**C**) Quantitation of results showing time of onset of depolarization (top) and duration of depolarization (bottom) in CCAP neurons from TN3 and AN1-4 (neurons in AN8 and AN9 could not be visualized). (**D**) Comparison between timecourse of GCaMP and ArcLight signal. Data in (**C,D**) are mean ± SEM. N = 10–13 for all genotypes. In (**C**), statistically significant differences (p<0.05) relative to control are indicated with '*'; *t* test (unpaired, two-tailed). In (**D**), statistically significant differences (p<0.05) are indicated by different letters; one-way ANOVA, Tukey's *post-hoc* multiple comparison analyses. Actual p values for all analyses can be found in **Supplementary file 1**. Individual measurements have been superimposed on the summary histogram when <8 neurons showed a measurable response. Genotype of animals expressing ArcLight sensor: CCAP>Arclight; of those expressing GCaMP: CCAP>GCaMP.

arrowheads in *Figure 9B*), one of them (top red record in *Figure 9B*) then showed a series of spikes, whereas the other (top blue record in *Figure 9B*) showed a more sustained response. We assume that the neuron expressing the spikes corresponds to the α neuron (*Figure 3D*), whereas the other corresponds to the β member of the pair. Most importantly, the main effect of challenging the preparations with ETH in the presence of picrotoxin was a shortening of the latency to depolarize, without significantly affecting the total duration of the entire response (*Figure 9C*). Finally, a quantitative analysis of the latency and duration of the voltage vs. GCaMP response (*Figure 9D*) reveals that both responses show an overall very similar timecourse. These results suggest that GABA inhibition delays the depolarization of CCAP neurons caused by ETH. This inhibition would then be lifted at around 20 min, causing the depolarization and firing of CCAP neurons, which results in the calcium spikes recorded using the GCaMP sensor (*Figure 8C*) and the activation of the ecdysis motor program.

## Discussion

Ecdysis behavior consists of behavioral routines and physiological events that are expressed in a specific sequence. At the neural level, the sequential nature of ecdysis is based on the sequential activation of different ETH targets. Here, we have identified some of the elements involved in determining the time course of activation of CCAP neurons. The CCAP neurons express ETHR, and by targeting ETHR RNAi to these neurons we showed that the timecourse and intensity of their response to ETH is sensitive to the dosage of ETHR. The associated lengthening of pre-ecdysis and failures in ecdysis underscore the well-established role for these neurons as key regulators of ecdysis (*Kim et al., 2006*, *2015*; *Lahr et al., 2012*; *Park et al., 2003*). In addition to ETH, however, we show that several elements downstream of this triggering hormone including EH, PBURS, and GABA play a key role in determining the response of CCAP neurons (whether GABA neurons mediating this inhibition are direct or indirect targets of ETH is currently unknown). Thus, we found that the response of CCAP neurons is qualitatively changed in the absence of EH, causing a decrease in the amplitude and dramatically altering the temporal features of the response, with a concomitant failure to ecdyse. Interestingly, we show that PBURS, which is produced by a subset of CCAP neurons, is also involved in patterning the response of these neurons to ETH, revealing a (direct or indirect) autocrine regulation of CCAP activation. Finally, we show that GABA inhibition mediates at least part of the delay between exposure to ETH and CCAP neuron activation and may also sculpt the shape of the resulting pattern of activation. Our findings are summarized in the model shown in *Figure 10*; the model details the contributions to the activation of CCAP neurons from ETH, peptides produced by targets of ETH including the CCAP neurons themselves, as well as inhibitory effects mediated by GABA, and define the times when these influences participate during the expression of the different phases of ecdysis. In addition to these actions, there are several features of the response to ETH whose origins await elucidation. For instance, decreasing ETH effectiveness using RNAi or different concentrations of ETH revealed the existence of non-linear processes whose basis is currently unknown. They could simply reflect the non-linear activation of GPCRs and/or may occur because the various targets that are activated or inhibited by ETH may respond differentially and non-linearly to different concentrations of this neuropeptide, as occurs in other cases of neuromodulation (*Brezina, 2010*; *Marder et al., 2014*). Similarly, there remains a 10-min delay in the onset of the response of CCAP neurons that is not mediated by GABA inhibition. Dissecting the contributions of different modulators and their targets to the response of this neuronal ensemble will undoubtedly be aided by the availability of recently developed genetic tools (*Diao et al., 2015*; *Luan et al., 2006*) that can be used for the precise manipulation of receptor spatial expression. Finally, we still do not know how the activation of the various peptidergic targets of ETH causes the production of the ecdysial motor programs (cf., *Figure 1A,B*). Preliminary evidence indicates that CCAP may play an important role in directly activating these motor programs because the timing of GCaMP activity in motoneurons follows closely the pattern of GCaMP activity of CCAP neurons (Mena and Ewer, unpublished).

In many motor systems, repetitive motor outputs are produced by central pattern generators (CPG's) whose exact pattern of activity is then modulated by biogenic amines and neuropeptides (*Brezina, 2010*; *Marder et al., 2014*; *Nusbaum and Blitz, 2012*). In the case of the ecdysis motor program, intrinsic modulatory actions (*Katz, 1995*) appear to play a defining role in the expression of the neural response elicited by ETH and of its accompanying motor output. Thus, for example,

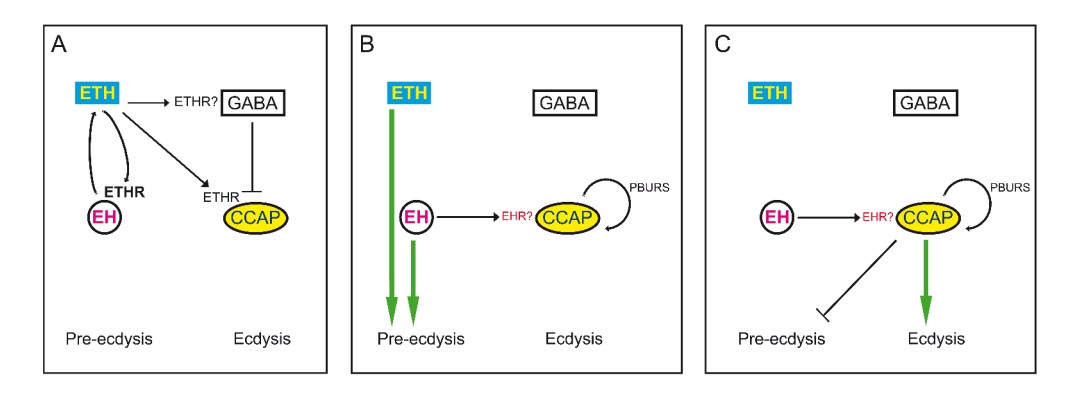

**Figure 10.** Model for endocrine control of ecdysial behaviors. (**A**) ETH released from peripheral endocrine cells acts directly on EH and CCAP neurons, and directly or indirectly on GABA neurons. Release of ETH is further potentiated by ETH-induced EH release (reciprocal arrows). Direct GABA inhibition of CCAP neurons prevents onset of response. (**B**) EH and/or ETH turn on the preparatory phase of ecdysis; waning of GABA inhibition, and EH and autocrine PBURS action (presumably mediated by BURS+PBURS bursicon heterodimer) on CCAP neurons activates CCAP neurons. (**C**) Activation of CCAP neurons causes expression of ecdysis proper and silencing of pre-ecdysis phase. Except for reciprocal relationship between EH and ETH, and ETH actions on CCAP neurons, none of the actions indicated are known to be direct. Arrows indicate stimulation; cross bars indicate inhibition.

the lack of EH destroys the repetitive pattern of firing of CCAP neurons induced by ETH (*Figure 5D*) as well as that of the resulting motor output (*Figure 5A,B and F*). What might the advantage be of not hardwiring the neural bases of such a vital behavior as ecdysis? One clue may lie in the fact that, despite the diversity in ecdysial behaviors expressed by different arthropods during different molts, the neuropeptides that drive ecdysis as well as their receptors are extremely well conserved, and clear homologs can be identified even in chelicerates (*Christie et al., 2011*; *Grbić et al., 2011*; *Veenstra et al., 2012*), which diverged from the insects ca. 600 million years ago (*Regier et al., 2005*, *2010*). One way for such a conserved signaling pathway to produce different behavioral out-puts would be to change the pattern of receptor expression. Consistent with this hypothesis, the exact function of particular ecdysial neuropeptides can differ in different insect groups and stages (*White and Ewer, 2014*). For instance, whereas *Drosophila* lacking CCAP express normal ecdysis behaviors (*Lahr et al., 2012*), RNAi inhibition of CCAP causes ecdysial failures in *Tribolium* (*Arakane et al., 2008*). Likewise, bursicon is involved in postecdysial maturation following the emergence of the adult fly (*Honegger et al., 2008*), yet is involved in the control of the earlier ecdysial phase at pupation (*Lahr et al., 2012*). Finally, it is especially interesting to identify an inhibitory input to CCAP neurons mediated by GABA, as it could provide a route through which sensory input could modulate the time of onset of the ecdysial phase. Such an input appears to be absent in *Drosophila* larval and pupal ecdysis but plays an important role in orthopteran ecdysis (*Carlson and Bentley, 1977*; *Ewer and Reynolds, 2002*) as well as in dipteran (*Baker et al., 1999*) and lepidopteran adult emergence (*Ewer and Truman, 1997*). It will be very interesting to determine the extent to which differences in neuropeptide receptor expression underlie the diversity of ecdysial behaviors expressed by insects with different body plans. In other systems, the different responses elicited by well-conserved neuropeptides may also be accomplished through changes in receptor expression. Such appears to be the case for oxytocin in voles, where differences in pair bonding in monogamous vs. gregarious species are at least in part due to differences in receptor distribution (*Young et al., 1999*).

Neuropeptides can be released at synapses, along axons, and even from dendrites, from where they can influence distant targets with a timecourse that can span from seconds to hours (*Leng and Ludwig, 2008*; *van den Pol, 2012*). Despite this mode of delivery, with low spatial and temporal specificity, their actions are typically very specific and can lead to the expression of tightly regulated responses. In the case of ecdysis behavior, for example, EH and CCAP are both broadly released into the ventral CNS, yet the timing of pre-ecdysis and ecdysis is normally extremely precise

(*Figure 2*). Similarly, in the case of the mammalian circadian system, robust and precise circadian rhythmicity of pacemaker activity in the suprachiasmatic nucleus (SCN) and of the behavioral output depends on Vasoactive Intestinal Peptide (VIP), which is released in a broad paracrine manner in the SCN (*Hastings et al., 2014*). A similar situation may occur in the *Drosophila* circadian system, where the neuropeptide, Pigment Dispersing Factor (PDF), is critical for the coupling of the different circadian oscillators (*Lin et al., 2004*; *Shafer and Yao, 2014*; *Yao and Shafer, 2014*; *Yoshii et al., 2009*). Overall, these examples show that neuropeptides play a critical role in determining the precise output of a neural network and of the behavior it controls. In the case of ETH, the precision of the timing and phasing of the different ecdysial phases depends on its combinatorial actions with other neuropeptides including EH and CCAP, and which are likely to be highly non-linear (*Brezina, 2010*; *Marder et al., 2014*). A similar situation applies for other neuropeptide controlled behaviors such as hunger, where neurons in the hypothalamic arcuate nucleus that express Agouti-related protein (Agrp) regulate feeding by integrating inputs mediated by circulating hormones (ghrelin and letpin), transmitters (glutamate, and GABA) and other peptidergic neurons (POMC neurons) (*Atasoy et al., 2012*; *Gao and Horvath, 2007*; *Sohn et al., 2013*). These inputs provide a readout of the animal's physiological state and are then translated into a non-linear switch-like behavior, consistent with the switch-like features of hunger states (*Yang et al., 2011*). For most systems, however, we lack information on the detailed actions exerted by each neuropeptide within a network. The inner working of the peptidergic network that controls ecdysis provides a tractable system for understanding how neuropeptides trigger and modulate complex patterns of neuronal activity and behaviors.

## Materials and methods

### Fly strains and genetics

Flies stocks were maintained at room temperature (22–25°C) on standard agar/cornmeal/yeast media. Unless indicated otherwise, all strains were obtained from the *Drosophila* Bloomington stock center (Bloomington, Indiana, USA). The following GAL4 drivers were used; expression and reference are indicated in parenthesis: *Ccap-GAL4* (CCAP neurons; *Park et al., 2003*); *Eh-GAL4* (EH neurons; *McNabb et al., 1997*); *FMRFa-GAL4* (subset of FMRFa neurons; *Suster et al., 2003*); *C164-GAL4* (motoneurons; *Torroja et al., 1999*). RNAi lines were obtained from Vienna *Drosophila* RNAi Center (VDRC; Vienna, Austria) or the National Institute of Genetics (NIG; Shizuoka, Japan): UAS-*ETHR* RNAi (VDRC #42717); UAS-*pburs* RNAi (NIG 15284 R-1). In all experiments using RNAi, its effectiveness was boosted by including a UAS-*dcr2* transgene. The null allele of *pburs* is described in *Lahr et al. (2012)*; the null allele of *Eh* and an *Eh* rescue transgene are described in *Krüger et al. (2015)*. UAS-GCaMP 3.2 was kindly provided by J. Simpson (HHMI, Janelia farms, USA).

Homozygous null animals were always heterozygous for a null allele in combination with a genetic deletion that included the relevant gene (*Df(2)Exel6036* for *pburs*; *Lahr et al. (2012)* and *Df(3)Eh[-]* for *Eh*; *Krüger et al., 2015*). Stocks with homozygous lethal mutations were maintained heterozygous with *actin-GFP* expressing balancer chromosomes. All UAS-transgene bearing flies were crossed with wild-type animals to create heterozygous controls.

### *Ccap*-LexA

The promoter region from −512 to +44 from the Ccap transcription start was obtained by PCR from a Y17 vector containing DNA immediately 5' of the Ccap transcription start (*Park et al., 2003*); kindly provided by Benjamin White) using the forward Gateway primer 5'-GGGGACAAGTTTG TACAAAAAAGCAGGCTAGAGAGTCGCCTCGAAATTGCCTC and a reverse primer with (5´-GGGGAC AACTTTTGTATACAAAGTTGTGATCCACTTAGCAACCGACGCGC-3´) or without a minimal hsp-promotor sequence (5´- GGGGACCACTTTGTACAAGAAAGCTGGGTGATCCACTTAG-CAACCGACGCGC-3´). The PCR products were cloned into the Gateway pCasper-W-LexA::GAD vector (*Diegelmann et al., 2008*) and their sequence confirmed. The recombinant Ccap-promoter-LexA::GAD fusion construct (Ccap-LexA) was introduced into the germline of w[1118] flies using standard methods by BestGene (Chino Hills, CA). Several independent transformant lines were obtained.

### *pburs* rescue transgene

Flies bearing a *pburs* rescue transgene were created by transforming flies carrying a suitable ΦC31 'landing pad' on chromosome III with PacMan clone CH322153L1 (*Venken et al., 2009*). This clone is approximately 19.8 Kb in length and is centered on the *pburs* gene. Transformation was carried by Best Gene (Chino Hills, California, USA).

## Behavioral analyses

### Intact puparium preparation

Animals that had recently pupariated were examined and those containing a bubble in the mid-region of the puparium (late stage p4(i); *Bainbridge and Bownes, 1981*) were selected. Preliminary characterization of pupal ecdysis behavior was carried out using intact pupae, as described in *Lahr et al. (2012)*. Briefly, the pupae were selected and placed on their side on a microscope slide, and filmed at room temperature (20–22°C) under dim transmitted light using a Leica DMLB microscope (10 X magnification). One experimental and one control animal was filmed simultaneously.

### Puparium-free preparation

Detailed characterization of ecdysis behavior was done using puparium-free preparations, as described in *Kim et al. (2006)*. Briefly, immediately after the first signs of onset of ecdysial behaviors the pupa was surgically removed from the puparium in a drop of PBS (137 mM NaCl, 2,7 mM KCl, 10 mM $Na_2HPO_4$ and 2 mM $KH_2PO_4$, pH 7.3), and placed in a recording chamber filled with halocarbon oil (Sigma-Aldrich Chemical Co., MO) in order to prevent desiccation. The animals were filmed at room temperature (20–22°C) under dim transmitted light using a Leica DMLB microscope (20 X magnification).

## Imaging of $Ca^{2+}$ and voltage dynamics

Imaging of $Ca^{2+}$ dynamics was carried out essentially as described in *Kim et al. (2006)*. Briefly, animals containing a bubble in the mid-region of the puparium (late stage p4(i); *Bainbridge and Bownes, 1981*) were selected. They were then further staged to be within 4 hr before onset of pupal ecdysis. The animals were then dissected under cold PBS (137 mM NaCl, 2.7 mM KCl, 10 mM $Na_2HPO_4$ and 2 mM $KH_2PO_4$, pH 7,3), placed on the surface of 200 µl, 1.5% low melting temperature agarose solution (Sigma type VII-A; Sigma-Aldrich Chemical Co., MO), which was then left to harden for 30 min in a humidified chamber kept at 10–15°C. Preparations were then covered with Schneider´s Insect Medium (Sigma-Aldrich Chemical Co., MO) and imaged under an Olympus DSU Spinning Disc microscope (Olympus Corporation, Shinjuku-ku, Tokyo, Japan) using a 20 X W NA 0.50 or 40 X W NA 0.80 immersion lens. Fluorescent images were acquired using an ORCA IR2 Hamamatsu camera (Hamamatsu Photonics, Higashi-ku, Hamamatsu City, Japan) using the CellR Olympus Imaging Software (Olympus Soft Imaging Solutions, Munich, Germany). Preparations were first imaged for 5 min (exposures taken every 5 s) and those showing spontaneous activity (ca. 5% of the preparations) were discarded. They were then stimulated with synthetic ETH1 (Bachem Co., USA; referred to here simply as ETH) and GFP fluorescence captured every 5 s for 30, 60, or 90 min for the $Ca^{2+}$ sensor, GCaMP, and every 2 s for 60 min for the voltage sensor, ArcLight. When used, antagonists for GABA receptor A, Picrotoxin (Sigma-Aldrich Chemical Co., MO) or Gabazine (SR95531; Tocris Bioscience, Bristol, UK), were added 10 min prior to ETH1 challenge.

## Analysis of GCaMP and voltage intensity timecourses

Recordings were analyzed using CellR Olympus Imaging Software (Version 2.6) and fluorescence intensity calculated as ΔF/F. The data were further processed with Excel (Microsoft, WA); statistical analyses were carried out using Prism 6.0 (Graphpad Software Inc, CA). In general, only 50–75% of CCAP neurons were in focus in a given preparation and could therefore be quantitated. However, since for most genotypes and manipulations 90–100% of neurons showed a measurable response, each preparation yielded at least one independent measurement per segment. Exceptions to this were neurons in segment T3, where only one pair of neurons showed a measurable signal; and for some genotypes and manipulations, where many neurons failed to respond (e.g. preparations challenged with 60 nM ETH1; cf. *Figure 4*). In cases where <8 measurements were obtained, individual data points have been superimposed on the relevant summary histograms (*Figures 4, 5,* and *9*).

## Acknowledgements

We thank Julie Simpson (Janelia Farm, USA), Paul Taghert (Washington University, St Louis, USA) for flies, and Benjamin White (National Institute of Mental Health, National Institutes of Health, Bethesda, USA) for clone containing regulatory region of *Ccap* gene. We also thank Eve Marder for comments on the manuscript.

## Additional information

### Funding

| Funder | Grant reference number | Author |
|---|---|---|
| Fondo Nacional de Desarrollo Científico y Tecnológico | 1141278 | Wilson Mena John Ewer |
| Millenium Institute Grant | P09-022F | Wilson Mena John Ewer |

The funders had no role in study design, data collection and interpretation, or the decision to submit the work for publication.

### Author contributions

WM, JE, Conception and design, Acquisition of data, Analysis and interpretation of data, Drafting or revising the article; SD, Provided unpublished Ccap-LexA line, Contributed unpublished essential data or reagents; CW, Drafting or revising the article, Contributed unpublished essential data or reagents

### Author ORCIDs

Christian Wegener, http://orcid.org/0000-0003-4481-3567
John Ewer, http://orcid.org/0000-0002-6806-3628

## Additional files

### Supplementary files

• Supplementary file 1. Actual p values for statistical analyses for results reported in main *Figures 2– 9*, and for *Figure 3—figure supplement 1*.

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
