## [Decision Letter]

Thank you for submitting your article "Stereotyped responses of *Drosophila* peptidergic neuronal ensemble depend on downstream neuromodulators" for consideration by *eLife*. Your article has been favorably evaluated by Eve Marder (Senior Editor) and three reviewers, one of whom, Ronald L Calabrese (Reviewer #1), is a member of our Board of Reviewing Editors. The following individuals involved in review of your submission have agreed to reveal their identity: Orie T Shafer (Reviewer #2) and Hans-Willi Honegger (Reviewer #3).

The reviewers have discussed the reviews with one another and the Reviewing Editor has drafted this decision to help you prepare a revised submission.

Summary:

This study combines the use of genetically encoded calcium- and voltage-sensitive probes, targeted RNAi expression, mutants null for particular neuropeptides downstream of ETH, and pharmacology, to understand how pupal ecdysis is orchestrated in *Drosophila*. Focusing on a subset of neurons that produce CCAP (Crustacean cardioactive peptide), which known play a key role in the control of ecdysis, they show that the response of CCAP neurons to ETH and the ensuing ecdysis behaviors depend on direct actions mediated by ETH as well as on the actions of neuropeptides downstream of ETH together with inhibition mediated by GABA. Removal of a downstream neuropeptide, EH, can eliminate the rhythmic pattern of neuronal activity induced by ETH. These results indicate that neuropeptides are critical for the expression of the fundamental features, including timing of the underlying neuronal program for pupal ecdysis. The work has broad implication for how neuropeptides can orchestrate complex behaviors and imply that they build flexibility into neuronal networks that can be exploited developmentally and evolutionarily.

Essential revisions:

There are a few concerns about controls (detailed in reviewer #2 comments) and also some concerns about the complexity of the figures, particularly Figure 4. The manuscript focuses on two major questions: 1) Which mechanisms are responsible for producing the sequential activation of neuronal targets in response to a common ETH stimulus (latency responses), and 2) How does the amplitude of the CCAP response (Ca+ fluorescence) relate to the ecdysis motor pattern. These questions may help in the reorganization of the Figures. The authors may break up the results of their experimental approach in the figures more clearly with respect to answering these questions. The major concerns of reviewers #2 and #3 are listed below to guide in your revision.

Minor points:

See minor revisions suggested by each reviewer (below).

*Reviewer #1:*

Explain the statistical differences more fully in the text. Give exact p values.

*Reviewer #2:*

There are two concerns that, if addressed, would improve this important study.

First, some critical experimental details are not adequately described in the manuscript. For example, the genotypes of the flies used in imaging experiments on RNAi manipulated brains should be clearly indicated either in the Results section or figure legend for every experiment. For example, it isn't clear from the description of the imaging experiments in Figure 2 how GCaMP and the ETH-R RNAi constructs were independently driven in motor neurons and CCAP expressing neurons, respectively. Was the CCAP-LexA used to drive UAS-RNAi? If so, there does not seem to be a LexAop-ETH-RNAi element described. Or was a LexA used for motor neurons (in which case the driver and the LexAop-GCaMP should be described)? With the genotypes unclear, it is difficult to access how well controlled these experiments are. For example, how well have the authors controlled for the possibility of leaky RNAi expression in the imaging experiments? Some evidence that the genetic driver is required for the RNAi effects would strengthen these experiments. The authors should ensure that every fly line used in the study is described in the Methods section.

Second, a reorganization of the figures would make this excellent manuscript more easily read. As organized, there are several highly related experiments that are spread over two or even three figures in the manuscript. The need to move between multiple figures to compare behavioral and physiological data for similar experimental manipulations makes the manuscript difficult to follow at times. For example, Figure 4 does a nice job summarizing a great deal of data, but as presented it is a bit challenging to navigate. I would consider breaking these data up among the other figures. The reader would be better served if the ETH-R knockdown data in Figure 4 were included in the current Figure 3 and if the ETH dose response data were combined with the current Figure 5. Likewise, the authors might break up the comprehensive behavioral data shown in Figure 1, placing the various genetic manipulations alongside the imaging done on the same manipulations. This would allow the reader to move more easily from one figure to the next while reading the Results section.

*Reviewer #3:*

There are a few concerns that should be addressed:

1) The figures indicate that the response of CCAP neurons in T3 to the diverse experimental manipulations differ from those in the abdominal neuromers 1-4 (AN1-4), see Figure 3, # of spikes; Figure 4, amplitude of GCaMP response; Figure 5 and Figure 8, # of spikes; Figure 9, duration. These differences deserve an explanation.

2) Also, AN 8 and 9 responses are different from those in AN 1-4, especially as shown in Figure 7. The authors mention that AN8-9 may not express ETHR but may also lack the PBURS (BURS) receptors.

3) I have problems with the notion that it is PBURS that plays a role in the control of ecdysis behaviors. It has been shown that the two subunits of bursicon (BURS and PBURS) heterodimerize spontaneously to the bioactive BURSICON if expressed together. Thus, the knock down of PBURS expression would prevent the production of a bioactive bursicon. In my opinion, it is simpler to assume that BURSICON has a feedback effect on the CCAP/BURSICON expressing AN neurons. Otherwise, the authors must show that in the early phases of ecdysis only PBURS is expressed, not BURS, and need to hypothesize a specific receptor for PBURS in these cells.

4) The results indicate that not the entire GABA inhibition acts directly on the CCAP neurons but some input must be indirect. The authors also cite publications indicating that inhibitory influences may originate from the brain and/or sub-esophageal ganglion. I would like to emphasize that the Benjamin White lab has shown convincingly that two bursicon-expressing neurons in the SEG project into all abdominal neuromers and thus may have a direct effect on the release of bursicon from the ANs. It might be worthwhile to discuss whether the indirect GABA inhibition on these cells originates from the inhibition of the SEG neurons.

5) The results provide convincing support that the peptidergic modulation of the CCAP neurons plays a critical role in the control of *Drosophila* ecdysis behavior. However, the ecdysis motor program is controlled by motor neurons and the authors show that the motor program as well as the alternating motor neuron activity was also severely disrupted in ETH RNAi puparium-free preparations. The authors should discuss the possible connections translating the activity of the AN-CCAP cells into the motor neuron pattern controlling the ecdysis.

6) I would suggest using * throughout to indicate statistically significant differences in the figures.

---

## [Author Response]

*Essential revisions:*

*There are a few concerns about controls (detailed in reviewer #2 comments) and also some concerns about the complexity of the figures, particularly Figure 4. The manuscript focuses on two major questions: 1) Which mechanisms are responsible for producing the sequential activation of neuronal targets in response to a common ETH stimulus (latency responses), and 2) How does the amplitude of the CCAP response (Ca+ fluorescence) relate to the ecdysis motor pattern. These questions may help in the reorganization of the Figures. The authors may break up the results of their experimental approach in the figures more clearly with respect to answering these questions. The major concerns of reviewers #2 and #3 are listed below to guide in your revision.*

We agree with the “Major questions” identified above; however, since most of the manipulations we performed affected both the temporal organization of the response to ETH and its amplitude it would not be possible to present these 2 issues separately. Thus, we have not altered significantly the order of presentation of the results. However, we have followed the reviewers’ advice and reorganized the figures to make life easier for the reader. A detail point-by-point response is provided below.

*Reviewer #1:*

*Explain the statistical differences more fully in the text. Give exact p values.*

We have provided a more detailed description of the statistical differences in the legends of the relevant figures (Figure 2–Figure 9). Since there are so many comparisons we have not indicated the exact p values for each one; rather we have included a supplementary table that includes the exact p values for all statistical analyses performed ([Supplementary-material SD1-data]).

*Reviewer #2:*

*There are two concerns that, if addressed, would improve this important study.*

*First, some critical experimental details are not adequately described in the manuscript. For example, the genotypes of the flies used in imaging experiments on RNAi manipulated brains should be clearly indicated either in the Results section or figure legend for every experiment. For example, it isn't clear from the description of the imaging experiments in Figure 2 how GCaMP and the ETH-R RNAi constructs were independently driven in motor neurons and CCAP expressing neurons, respectively. Was the CCAP-LexA used to drive UAS-RNAi? If so, there does not seem to be a LexAop-ETH-RNAi element described. Or was a LexA used for motor neurons (in which case the driver and the LexAop-GCaMP should be described)? With the genotypes unclear, it is difficult to access how well controlled these experiments are. For example, how well have the authors controlled for the possibility of leaky RNAi expression in the imaging experiments? Some evidence that the genetic driver is required for the RNAi effects would strengthen these experiments. The authors should ensure that every fly line used in the study is described in the Methods section.*

We apologize for omitting these details. A description of the exact genotypes has now been added to the legends of the data figures (Figure 1–Figure 8 and Figure 3—figure supplement 1). Specifically with respect to Figure 1 (= ex-Figure 2), motoneurons (MN) were imaged while knocking down ETH-R RNAi in CCAP neurons by driving ETH-R RNAi (and GCaMP) using a combined CCAP-GAL4+MN-GAL4 driver. The response of CCAP neurons is clearly distinguishable from MN activity. In addition, expressing ETH-R RNAi only in MN’s had only a minor effect on ecdysis, affecting only the frequency of pre-ecdysial contractions, measured in intact puparium preparations (Figure 1). Thus, the GCaMP signal recorded using the CCAP-GAL4+MN-GAL4 driver reflects primarily (if not exclusively) the effects of knocking down ETH-R in CCAP neurons. This issue has been clarified in the legend to Figure 1 (= ex-Figure 2).

With respect to possible leakage from UAS-RNAi constructs, we always tested (and reported) the relevant controls and did not detect any signs of leakage (Figure 2, for UAS-ETHR RNAi; Figure 8, UAS-GABA-RA, for GABA-RA). We have double-checked the Methods section to make sure it includes a description of all the fly line used in this study.

*Second, a reorganization of the figures would make this excellent manuscript more easily read. As organized, there are several highly related experiments that are spread over two or even three figures in the manuscript. The need to move between multiple figures to compare behavioral and physiological data for similar experimental manipulations makes the manuscript difficult to follow at times. For example, Figure 4 does a nice job summarizing a great deal of data, but as presented it is a bit challenging to navigate. I would consider breaking these data up among the other figures. The reader would be better served if the ETH-R knockdown data in Figure 4 were included in the current Figure 3 and if the ETH dose response data were combined with the current Figure 5.*

We agree with this suggestion and have attempted to improve the readability of the manuscript. For this we have:

- Inverted the order of the first two figures.

- Included in a single figure the behavioral results obtained from intact puparium and puparium free preparations following a given manipulation; and, whenever space allowed it, these results have been presented together with the corresponding GCaMP results. Thus, Figure 2 (ex-Figure 1) now includes both types of behavioral results obtained following ETH-R knockdown; Figure 5 now includes both types of behavioral results obtained in the absence of EH function as well as the corresponding GCaMP activity of CCAP neurons in response to ETH; Figure 6 includes both types of behavioral results obtained in *pburs* mutants, followed by Figure 7, which shows the responses of CCAP neurons in a *pburs* mutant background. Finally, Figure 8 shows both types of behavioral results obtained when GABA-RA was knocked down in CCAP neurons as well as the corresponding GCaMP activity of the CCAP neurons in response to ETH.

- We have also eliminated Figure 4 as a main figure and added the relevant information in the text. We chose this option rather than breaking up Figure 4 and adding parts of it to the relevant figures because for each genotype/manipulation we only report 2 pieces of information: the amplitude of the GCaMP response in CCAP neurons from segments TN3 and AN1-4 relative to control. This information has now been added to the relevant section of the Results (subsection “Role of ETH signaling in the activation of downstream peptidergic targets”, third and fourth paragraphs; subsection “Influence of downstream peptides on the pattern of activation of CCAP neurons by ETH: role of Eclosion hormone”, second paragraph and subsection “Influence of downstream peptides on the pattern of activation of ETH targets: role of PBURS”), and is therefore provided to the reader at the appropriate time. A similar approach was used to report the% responsiveness of CCAP neurons to ETH following different manipulations (e.g., subsection “Role of ETH signaling in the activation of downstream peptidergic targets”, fourth paragraph). The contents of Figure 4 has nevertheless been preserved but moved to Figure 3—figure supplement 1.

*Likewise, the authors might break up the comprehensive behavioral data shown in Figure 1, placing the various genetic manipulations alongside the imaging done on the same manipulations. This would allow the reader to move more easily from one figure to the next while reading the Results section.*

We have combined the behavioral results from intact puparium and puparium free preparations, as described above.

*Reviewer #3:*

There are a few concerns that should be addressed:

*1) The figures indicate that the response of CCAP neurons in T3 to the diverse experimental manipulations differ from those in the abdominal neuromers 1-4 (AN1-4), see Figure 3, # of spikes; Figure 4, amplitude of GCaMP response; Figure 5 and Figure 8, # of spikes; Figure 9, duration. These differences deserve an explanation.*

We agree with this reviewer that for several manipulations the response of TN3 neurons differs from that of abdominal neuromeres AN1-4. We don’t understand the basis for such differences, which could be due to different levels of expression of ETHR, different wiring, etc. We have added text highlighting these segmental differences (subsection “Role of ETH signaling in the activation of downstream peptidergic targets”, third paragraph).

*2) Also, AN 8 and 9 responses are different from those in AN 1-4, especially as shown in Figure 7. The authors mention that AN8-9 may not express ETHR but may also lack the PBURS (BURS) receptors.*

We agree that the responses of CCAP neurons in AN8 and AN9 differ from those seen in AN1-4. We do not know the basis for these differences but one factor that likely contributes is that the AN8 and AN9 CCAP neurons do nor express ETHR (Kim et al., 2006; Diao et al., 2016) and are thus insensitive to the direct actions of ETH. Yet, we appreciate the reviewer’s comment that they likely also do not express the bursicon receptor since eliminating PBURS did not affect the (indirect) response of these neurons to these downstream hormones. We have added text mentioning this possibility (subsection “Influence of downstream peptides on the pattern of activation of ETH targets: role of PBURS”).

*3) I have problems with the notion that it is PBURS that plays a role in the control of ecdysis behaviors. It has been shown that the two subunits of bursicon (BURS and PBURS) heterodimerize spontaneously to the bioactive BURSICON if expressed together. Thus, the knock down of PBURS expression would prevent the production of a bioactive bursicon. In my opinion, it is simpler to assume that BURSICON has a feedback effect on the CCAP/BURSICON expressing AN neurons. Otherwise, the authors must show that in the early phases of ecdysis only PBURS is expressed, not BURS, and need to hypothesize a specific receptor for PBURS in these cells.*

We agree with this reviewer that the bioassays used to measure the activity of BURS-BURS and PBURS-PBURS homodimers, and of BURS-PBURS heterodimers show that only the heterodimer (= bursicon) can activate the bursicon receptor in a heterologous assay and cause tanning in intact flies bioassay (Luo et al., 2005; Mendive et al., 2005). Nevertheless, there is the nagging fact that PBURS and BURS are not strictly co-expressed. Most notably, BURS is expressed in 1 pair of CCAP neurons in 8 abdominal segments whereas PBURS is only expressed in 4 of these. Thus, there are 4 segments in which CCAP neurons express only the BURS subunit of bursicon. Even though there is currently no activity associated with BURS-BURS homodimers it is difficult to escape the possibility that homodimers may be active in some context. Since we don’t know which this activity may be, we chose to use the more conservative description of our results because, in our manipulations, all that we are really sure of is that we eliminated PBURS. Nevertheless, we do acknowledge in the text that the lack of PBURS causes a lack of bursicon (=BURS-PBURS heterodimer) (“Consistent with the findings of (Lahr et al., 2012), the lack of PBURS (and **thus of the heteromeric bursicon**; our emphasis here) in intact puparia caused a significant lengthening of ecdysis, with only 5/8 animals executing the behavior (Figure 6).”

*4) The results indicate that not the entire GABA inhibition acts directly on the CCAP neurons but some input must be indirect. The authors also cite publications indicating that inhibitory influences may originate from the brain and/or sub-esophageal ganglion. I would like to emphasize that the Benjamin White lab has shown convincingly that two bursicon-expressing neurons in the SEG project into all abdominal neuromers and thus may have a direct effect on the release of bursicon from the ANs. It might be worthwhile to discuss whether the indirect GABA inhibition on these cells originates from the inhibition of the SEG neurons.*

We are well aware of these studies (Luan et al., 2006; and Peabody et al., 2009), yet they apply to the control of bursicon release and wing expansion, both of which take place *after* adult emergence (which corresponds to adult ecdysis); they are thus not relevant to pupal ecdysis, where genetic evidence indicates that bursicon plays a role in the *execution* of ecdysis itself (Lahr et al., 2012). Although the inhibitory SEG neurons described in these reports and mentioned by this reviewer could contribute the delay in expression of the ecdysial phase of ecdysis, there is no direct evidence that this is the case. We did not cite this work because the control of ecdysis differs between pupal and adult ecdysis (e.g., the role of bursicon changes dramatically). Thus we do not feel comfortable extrapolating to pupal ecdysis neuronal elements that control adult ecdysis, especially if they regulate events that are expressed *after* adult ecdysis.

Prior to our finding the only available information was that decapitation of the animal after ETH release significantly reduced the latency to ecdysis. However, given the radical nature of this manipulation it would be difficult to argue convincingly that this inhibition was mediated by an inhibition specifically originating from the brain or subesophageal ganglion. Here we have advanced our understanding of the nature of this inhibition by showing that at least part of the delay in activation of CCAP neurons is due to a direct inhibition of CCAP neurons by GABA. Future studies will attempt to identify the relevant GABA-ergic neurons, and determine whether they respond directly or indirectly to ETH.

*5) The results provide convincing support that the peptidergic modulation of the CCAP neurons plays a critical role in the control of Drosophila ecdysis behavior. However, the ecdysis motor program is controlled by motor neurons and the authors show that the motor program as well as the alternating motor neuron activity was also severely disrupted in ETH RNAi puparium-free preparations. The authors should discuss the possible connections translating the activity of the AN-CCAP cells into the motor neuron pattern controlling the ecdysis.*

Some of the players that transduce ETH actions within the CNS have been identified. They are neurons that express the “A” isoform of the ETH receptor (ETH-RA) and consist of peptidergic neurons that express kinin, FMRFamide, EH, CCAP (in combination with MIP and bursicon). In addition there are several dozen neurons that express ETH-RB, whose identity is currently unknown. Yet it is unlikely that motoneurons are relevant ETH targets. Indeed, expressing ETHR RNAi in motoneurons had, at most, a modest effect on ecdysis behavior: no effect was detected on ecdysis behavior monitored in intact preparations, and only an effect on ecdysis frequency could be detected in puparium free preparations (Figure 2). Thus it is unlikely that motoneurons are direct targets of ETH. Rather it is possible that peptide downstream of ETH are involved in directly controlling or modulating the ecdysis motor programs. Preliminary evidence indicates that CCAP may play an important role, because the timing of GCaMP activity in motoneurons follows closely the pattern of GCaMP activity of CCAP neurons. A sentence to this effect has been added to the Discussion (first paragraph).

*6) I would suggest using * throughout to indicate statistically significant differences in the figures.*

We appreciate this suggestion and have used “*” for comparisons that are between experimental vs. control (e.g., Figure 2, Figure 5, Figure 6, Figure 7, Figure 8 and Figure 9). However, for cases where we made multiple comparisons test we think it is more appropriate to use letters (a different letter for statistically different groups). The results of such comparisons can be complex and using “*” and brackets would unnecessarily clutter the figures.